# Extensive co-binding and rapid redistribution of NANOG and GATA6 during emergence of divergent lineages

Joyce J. Thompson[1], Daniel J. Lee[1], Apratim Mitra [2], Sarah Frail [1], Ryan K. Dale[2] & Pedro P. Rocha [1,3✉]

Fate-determining transcription factors (TFs) can promote lineage-restricted transcriptional programs from common progenitor states. The inner cell mass (ICM) of mouse blastocysts co-expresses the TFs NANOG and GATA6, which drive the bifurcation of the ICM into either the epiblast (Epi) or the primitive endoderm (PrE), respectively. Here, we induce GATA6 in embryonic stem cells–that also express NANOG–to characterize how a state of co-expression of opposing TFs resolves into divergent lineages. Surprisingly, we find that GATA6 and NANOG co-bind at the vast majority of Epi and PrE enhancers, a phenomenon we also observe in blastocysts. The co-bound state is followed by eviction and repression of Epi TFs, and quick remodeling of chromatin and enhancer-promoter contacts thus establishing the PrE lineage while repressing the Epi fate. We propose that co-binding of GATA6 and NANOG at shared enhancers maintains ICM plasticity and promotes the rapid establishment of Epi- and PrE-specific transcriptional programs.

[1] Unit on Genome Structure and Regulation, National Institute of Child Health and Human Development, National Institutes of Health, Bethesda, MD 20892, USA. [2] Bioinformatics and Scientific Programming Core, National Institute of Child Health and Human Development, National Institutes of Health, Bethesda, MD 20892, USA. [3] National Cancer Institute, NIH, Bethesda, MD 20892, USA. ✉email: pedrorocha@nih.gov

Recent single-cell (sc) studies have highlighted the prevalence of plastic states characterized by co-expression of divergent lineage-determining TFs during cell specification[1,2]. The mouse second cell fate decision is an in vivo paradigm to understand how lineage-specific TFs regulate differentiation of diverse cell types from common progenitors. In mice, Epiblast (Epi) and Primitive Endoderm (PrE) arise from a common progenitor in the early blastocyst, the inner cell mass (ICM). Around embryonic day 3.25 (E3.25), all ICM cells express heterogeneous levels of both Epi Transcription factors (TFs) –SOX2, OCT4, and NANOG– and PrE TF GATA6[3,4]. By E3.5, ICM cells restrict expression to either NANOG or GATA6, which predisposes them towards the Epi or PrE fate, respectively[4–8]. By E4.5, both lineages are established, specified PrE cells segregate to form an epithelial layer encapsulating the pluripotent Epiblast, and the blastocyst implants in the uterus, culminating pre-implantation development[4,9,10].

ICM cells preferentially expressing GATA6, and subsequently its downstream targets SOX17 and GATA4, activate a transcriptional network promoting commitment to the PrE fate[6,11–16]. However, PrE precursors in early blastocysts can still switch to an Epi fate if the ratio of Epi/PrE compartments is affected[17,18], indicating that cells at this stage are plastic and not fully committed. This adaptability is lost in the late blastocyst (E4.5) suggesting that Epi and PrE fates are irreversibly committed at this stage[18,19]. Although Epi- and PrE-specific genes have been described using scRNA-seq in developing blastocysts[20–22], we lack understanding of how plasticity is maintained in the ICM and then quickly lost as cells are specified into two distinct lineages.

Embryonic stem (ES) cells cultured in vitro do not express GATA6 or any other PrE-specific genes, and have been used to study regulatory mechanisms controlling the Epi pluripotency network[23–27]. Similarly, extra-embryonic endoderm (XEN) cells derived from blastocysts have been used to characterize the PrE fate in vitro. However, XEN cells are a heterogenous population, more closely resembling the PrE post-implantation derivatives, parietal, and visceral endoderm[28,29]. Alternatively, ectopic expression of PrE TFs like GATA6, GATA4 or SOX17 in ES cells can induce transdifferentiation into the PrE state and provide efficient temporally controllable systems, which have helped identify genes and signaling pathways driving PrE establishment[30–35].

In this study, we aim to simulate the plastic state of bipotent ICM cells. We use ES-cells carrying an inducible GATA6 transgene and profile them immediately following GATA6 induction (2 h). Although GATA6 induction leads to repression of NANOG and other Epi TFs, at this early timepoint both GATA6 and NANOG are highly expressed. We find that during this stage of co-expression, GATA6 binds its motifs in cis-regulatory elements (CREs) of PrE genes and works as a pioneer TF to make them accessible for transcriptional activation. Simultaneously, GATA6 binds at an unexpectedly large fraction of CREs controlling Epi genes, leading to co-occupancy with the core pluripotency TFs, NANOG and SOX2. Surprisingly, we observe that transient binding of GATA6 to Epi CREs is followed by eviction of NANOG and SOX2 and their redirection to GATA6-bound PrE CREs. The ability of GATA6 and NANOG to bind at both Epi and PrE loci is evident also in uncommitted ICM cells in blastocysts. We propose that co-binding of NANOG and GATA6 to the same regulatory elements confers plasticity in the ICM and allows rapid bifurcation into divergent lineages during blastocyst development.

## Results

**GATA6 expression quickly generates PrE-like cells in vitro**. To characterize how GATA6 and NANOG coregulate plasticity and

how GATA6 induces the PrE state while inhibiting the alternative Epi fate, we employed doxycycline (Dox)-induced GATA6 expression in ES cells[35]. GATA6 expression was induced for 12 h and cells cultured up to 96 h (Fig. 1a). Since ES cells in culture resemble Epi cells, Dox-induced GATA6 expression initiates transdifferentiation of Epi-like cells into PrE. To identify the transdifferentiation time points that resembled in vivo stages more closely, we used a scRNA-seq dataset[22] generated from E3.5 and E4.5 blastocysts and compared them to bulk RNA-seq of GATA6-induced cells at each time point (Fig. 1b). To achieve this, we first obtained normalized expression data using standard procedures specific to each data source (see Methods for details). Next, we fit a linear model to account for technical variability of the different platforms and performed principal component analysis (PCA) on the residuals to compare bulk and scRNA-seq timepoints. Reassuringly, trophectoderm (TE) cells, which are specified independently in the first cell fate decision, clustered away from all other data points. Each time point between 8 to 48 h after Dox induction, clustered with a distinct cell type in blastocysts. At 8 h, induced cells resembled ICM and Epi cells, while at 16 h they clustered with PrE precursors present in early blastocysts (E3.5). Between 24 and 48 h, in vitro cells resembled the PrE at E4.5, suggesting that the process of PrE fate-establishment had been completed. The 96 h time point clustered away from the blastocyst cells suggesting that by this time point cells most likely resembled PrE derivatives, deeming them less relevant to study PrE-specification mechanisms. FACS and immunofluorescence confirmed that as reported[35], almost all cells activate PrE targets and repress Epi genes by 48 h (Supplementary Fig. 1a, b), and immediately following Dox induction cells homogenously co-express GATA6 and NANOG (Supplementary Fig. 1c). We then looked at changes in expression of genes that form the Epi and PrE transcriptional networks. Epi- and PrE-specific genes were defined by identifying genes differentially expressed in the two ICM-derived lineages. For higher stringency, only genes identified as Epi or PrE-specific in a second dataset[21] were included in downstream analyses. Genes more highly expressed in the PrE population were included in the PrE transcriptional network (579 genes), and vice versa for Epi genes (221 genes). As shown in Fig. 1c, the transcriptome progressively changed during GATA6-induction, with a decrease of Epi-, and an increase of PrE-transcript levels. Differential gene expression analyses of the entire transcriptome also revealed progressive changes in expression, starting as early as 2 h post GATA6 induction, with 639 genes silenced, and 920 activated (Supplementary Fig. 1d, e). In summary, these data show that immediately following GATA6 induction, ES cells approach a transcriptional state that resembles the plastic ICM state at E3.5 and within 48 h, are more similar to PrE-specified cells at E4.5.

Although immunofluorescence and FACS support a homogenous PrE induction upon GATA6 expression, we wanted to confirm that this holds true for changes to the transcriptome. We compared bulk-RNA-seq in unsorted, and cells sorted for high PDGFRA expression, a PrE marker. The transcriptional profile of sorted cells was strikingly similar to that of unsorted cells (Supplementary Fig. 1f). Amongst the 130 and 343 genes that were differentially expressed between unsorted and sorted populations at the 8 and 24 h time points very few were part of the PrE-specific network (2 Epi genes at 8 h and 23 at 24 h, 1 PrE gene at 8 h and 8 at 24 h Supplementary Fig. 1g). However, even these genes showed similar expression changes during GATA6 induction (progressive activation of the 9 PrE genes, and repression of the 23 Epi genes), albeit with modestly different magnitudes (Supplementary Fig. 1h). Importantly, GATA6-induced expression changes of fate-determining PrE (*Gata4*, *Sox17*) and Epi (*Sox2*, *Nanog*) genes was identical between bulk

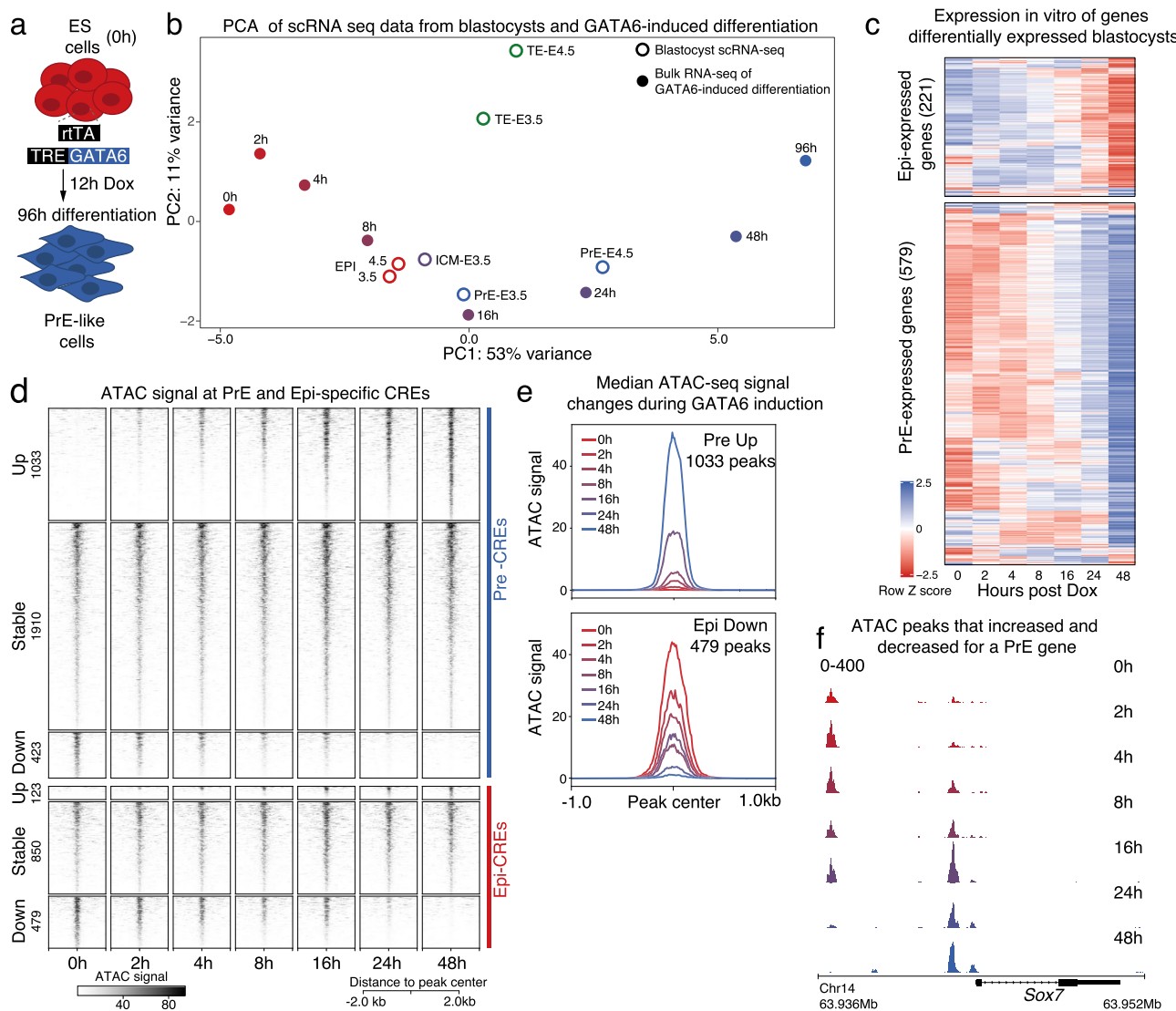

**Fig. 1 GATA6-driven in vitro differentiation recapitulates PrE specification in blastocysts and initiates rapid transcriptional changes and chromatin remodeling. a** Scheme depicting GATA6-driven in vitro differentiation of mES cells used to study PrE specification. **b** PCA of bulk-RNA-seq at differentiation time-points (solid circles) and single-cell RNA-seq profiles of E3.5 and E4.5 blastocysts (open circles). The PrE precursor state was achieved by 16 h of GATA6-driven differentiation and fully specified PrE-cells appeared at 24-48 h. **c** Heatmap of transcript z-scores of Epi- and PrE-specific genes shows that most blastocyst lineage-specific genes showed similar changes in expression during GATA6-driven differentiation. **d** Changes in chromatin accessibility measured by ATAC-seq at PrE and Epi CREs were quickly initiated within 2 h of differentiation. Accessible regions at 48 h were compared to 0 h and classified as up, stable or down. **e** Median ATAC signal plots depict an increase in ATAC signal at 1033 PrE-CREs and decrease at 479 Epi-Cres. **f** Browser view at the *Sox7* locus showing a progressive loss in ATAC signal at a distal CRE and a gain in accessibility at a proximal CRE. This example shows that not all PrE CREs gained accessibility during PrE-specification.

and sorted populations (Supplementary Fig. 1i). Even though we cannot completely rule out some cellular variability during GATA6 induction, these data suggest that majority of the cells induce a PrE fate very homogeneously.

**Chromatin remodels rapidly following GATA6 induction**. To probe into the dynamics of chromatin remodeling accompanying PrE establishment, we profiled changes in chromatin accessibility using ATAC-seq[36]. This revealed changes in chromatin accessibility by 2 h of GATA6 expression (Supplementary Fig. 1j–l) with little difference between PDGFRA[+]-sorted and bulk population (Supplementary Fig. 1m, n), and no single peak identified as differentially accessible between sorted and unsorted populations at any time point. This indicates that remodeling of chromatin landscape begins rapidly following GATA6 induction and with

little heterogeneity. As described in the methods, ATAC peaks were classified as Epi- or PrE-specific CREs using the closest TSS for a distance up to 50 kb, a method commonly used to annotate GWAS loci[37]. In both cases, the majority of CREs (1910) did not change accessibility during differentiation (Fig. 1d). Although most PrE CREs that showed changes were associated with an increase in accessibility (1033), some showed a reduction (423). Conversely, most dynamic Epi CREs lost accessibility (479) while few showed a gain (123). Gain of ATAC signal at PrE CREs was evident as early as 2 h with a surge at 16 h (Fig. 1e, top). At Epi CREs, loss of accessibility also began at 2 h and then decreased gradually (Fig. 1e, bottom). The 16 h surge in accessibility is exemplified at the promoter-proximal CRE controlling *Sox7*, a PrE-specific TF activated by GATA6 (Fig. 1f). In contrast, a distal *Sox7* CRE lost chromatin accessibility, which is a good example of

why not all PrE CREs gain accessibility as it may have been expected. Similar chromatin remodeling dynamics were observed at ATAC peaks identified throughout the genome, without restricting analysis to Epi or PrE-specific CREs (Supplementary Fig. 1k, l). These data demonstrate that GATA6-driven chromatin remodeling is initiated immediately following induction and that characterization of early events is crucial to understanding how a single transcription factor induces the PrE fate while inhibiting the Epi transcriptional program.

**GATA6 functions as a pioneer TF to rapidly remodel PrE loci.** To identify regulatory mechanisms contributing to changes in chromatin accessibility upon GATA6 induction, we mapped TF binding using CUT&RUN[38] and distribution of histone tail modifications associated with active and repressed chromatin using CUT&TAG[39]. We detected endogenous *Gata6* mRNA, as early as 2 h after addition of Dox, at levels comparable to those of the transgene (Fig. 2a) showing that GATA6 quickly auto-regulates its own expression. Other known GATA6 targets such as *Gata4, Sox17, Hnf1b* and *Pdgfra*, showed increased transcript levels only by 4–8 h (Fig. 2a). To understand how GATA6 and its targets regulate the PrE program, we mapped their binding during the course of differentiation. We first identified genome wide GATA6 peaks at early (2, 4, 8 h), and late (48 h) stages and by comparing with ATAC data, saw that GATA6 binding could be divided into three peak-types of similar proportions: early binding at regions that were already accessible before GATA6 induction, early binding at regions that only become accessible after induction, and regions bound only at late stages (48 h). These three GATA6-bound peak-types showed an increase in accessibility following binding and were detected both at PrE-specific CREs (Fig. 2b, c), and genome wide (Supplementary Fig. 2a, b). As defined for ATAC, GATA6 peaks were classified as PrE CREs if the closest TSS found within 50 kb of the peak was a PrE-specific gene. Deposition of H3K4me3, a mark of active transcription, increased at promoter-proximal PrE-CREs (within 5 kb) (Fig. 2b, Supplementary Fig. 2c), while H3K27ac, a mark of active enhancers, was deposited at promoter-proximal and -distal PrE-CREs (Fig. 2b, c). Irrespective of the timing of GATA6 binding (early or late), H3K27ac deposition trailed the gain of accessibility (Fig. 2b–d, Supplementary Fig. 2a) by a few hours likely reflecting the time required to recruit histone acetylation complexes following GATA6 binding. Together, this shows that PrE CREs bound by GATA6 are marked for activation leading to transcription of PrE-specific genes. Surprisingly, most CREs activated by GATA6 binding did not show enrichment of repressive histone marks (H3K27me3 and H3K9me3) at 0 h, in undifferentiated ES cells (Figs. 2b, c, example of a CRE with H3K27me3 at 0 h is shown in Supplementary Fig. 2b). We propose that this may contribute to the fast activation seen during PrE lineage commitment upon GATA6 induction, as these regulatory elements would not require removal of heterochromatin marks for activation.

While Dox induction led to immediate GATA6 binding to several CREs, others were bound only at 48 h (Fig. 2b, Supplementary Fig. 2a, cluster 3). To understand whether the number of GATA6 motifs within its target CREs could explain how GATA6 differentiates between its early and late target sites, we compared GATA6 motif density within these two types of CREs. Early GATA6 peaks had higher motif density per peak compared to late peaks (Supplementary Fig. 2a), which suggests that late peaks might require additional factors expressed later during differentiation to gain complete chromatin accessibility. Therefore, we assessed binding of GATA4 and SOX17 at late-differentiation stages. As expected, there was a large overlap in

binding targets for the three PrE TFs that was more prominent at the GATA6 sites bound only at 48 h (Supplementary Fig. 2e). In addition, we used HOMER motif enrichment at late GATA6 peaks by comparing with GATA6 early binding sites. The motif recognized by HNF1B, a TF shown to be important for visceral endoderm specification[40], was the most enriched motif (Supplementary Fig. 2e). Notably, ATAC-seq footprint protection–a measure of the likelihood of TF binding–showed mild protection of HNF1B motifs at 24 h, which increased by 48 h, coinciding with GATA6 binding at its late target sites (Supplementary Fig. 2f). This suggests that binding of HNF1B may aid GATA6 binding at late sites, along with GATA4 and SOX17. Notably, because of how these comparative analyses were performed, the GATA6 binding motif was not identified as enriched along with HNF1B, since GATA6 itself is bound at both early and late target sites.

We then asked whether GATA6, like other GATA-family members[41,42], functions as a pioneer TF capable of accessing its motifs even if occluded by nucleosomes. To answer this, we plotted ATAC-seq data representing nucleosomal occupancy by including fragments with length longer than a nucleosome (>150 bp). This filtered ATAC-seq signal was centered on the 6 bp GATA6 motif and plotted across early GATA6 peaks already accessible in uninduced cells (Fig. 2e, left panel), and early GATA6 peaks inaccessible in uninduced cells (Fig. 2e, right panel). In both instances, GATA6 motifs showed high nucleosomal occupancy in uninduced cells, which were quickly repositioned as early as 2 h post induction. This supports a role for GATA6 as a pioneer TF in activating the PrE gene regulatory network. As GATA6 affects nucleosome occupancy within 2 h of induction in vitro, it is likely that GATA6-expressing ICM cells, will have most of their PrE CREs already bound by GATA6, poising them for rapid differentiation towards the PrE fate. Moreover, GATA6 downstream transcription factors like GATA4 and SOX17, or other known chromatin remodelers only become upregulated by 4–8 h. This further supports a direct role of GATA6 in remodeling chromatin without indirect contribution from additional PrE-specific factors induced in response to the transgene expression. In fact, GATA6 can independently control chromatin accessibility also during definitive endoderm differentiation, further providing evidence for its pioneering ability[43]. In addition, we performed a comparative analysis similar to what we describe for HNF1B and found that as compared to 48h-specific GATA6 targets, the early GATA6-bound sites were enriched for motifs of the essential pluripotency factor NR5A2 (Supplementary Fig. 2e), known to control binding of NANOG/OCT4 and SOX2[44,45]. This points towards a potential contribution of GATA6 in repurposing pluripotency TFs for activation of the PrE program.

**Transient GATA6 binding precedes inactivation of Epi CREs.** In addition to activating the PrE transcriptional network, establishment of the PrE lineage requires repression of the alternative Epi fate. At the transcript level, the core Epi TFs, *Nanog* and *Sox2* were silenced progressively as induction of a PrE-like fate proceeded (Fig. 3a). As shown previously[46], levels of the pluripotency factor OCT4 remained stable (*Pou5f1*, green). Since the protein levels of NANOG, the Epi-determining TF, were completely depleted by 12 h (Supplementary Fig. 3a), we investigated how its genome-wide distribution changed during GATA6 induction only at early time points (0 to 8 h). As NANOG and SOX2 are known to bind and regulate self-renewal and pluripotency genes in ES cells[23], we also investigated how SOX2 genome-wide distribution changed during GATA6-induction.

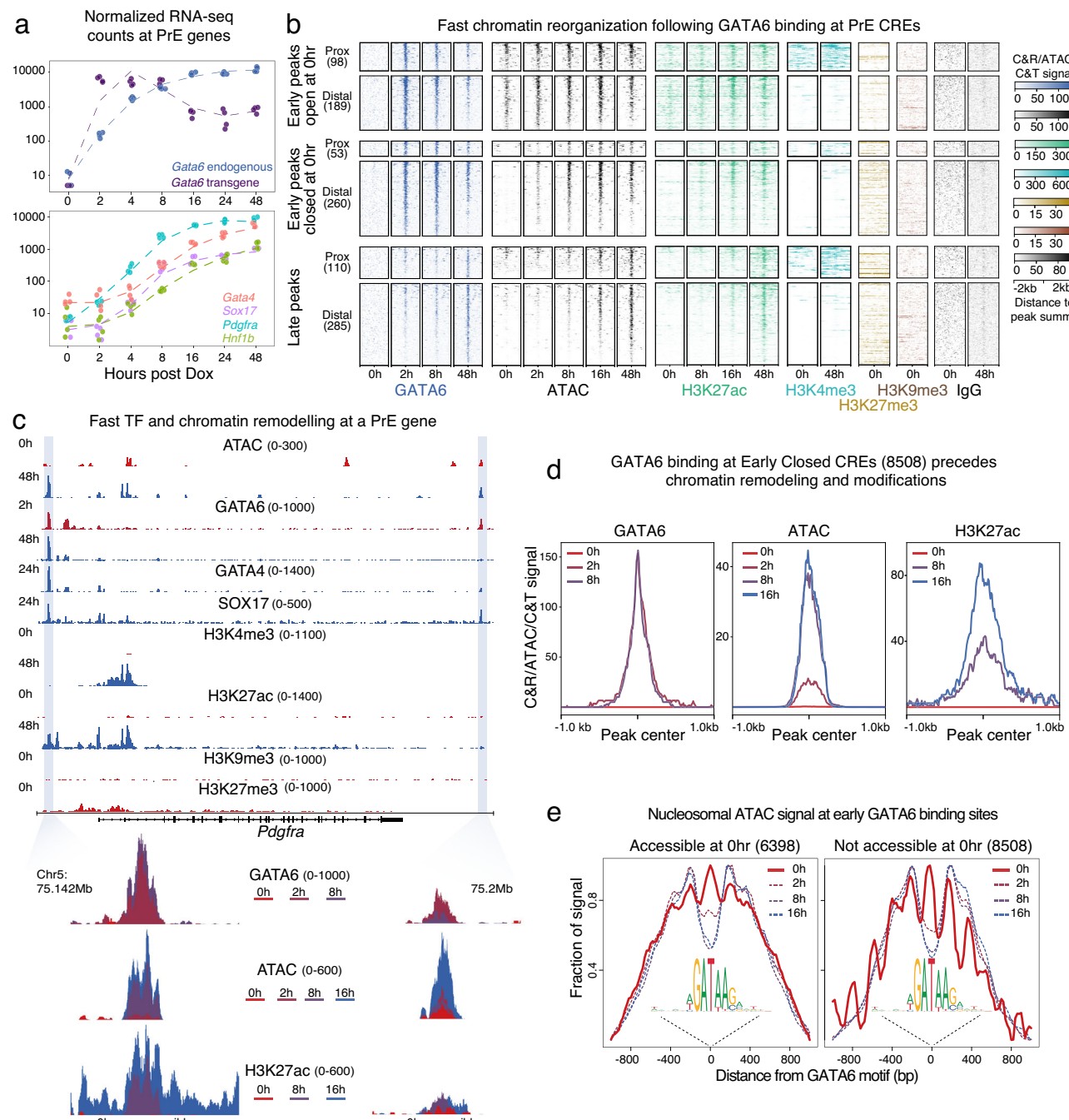

**Fig. 2 GATA6 binding at PrE-specific CREs precedes nucleosomal repositioning, changes in chromatin modifications and transcriptional activation of the PrE transcriptional program. a** Normalized RNA-seq counts show a gradual increase in endogenous *Gata6* levels that followed Dox-induced expression of the *Gata6* transgene (top panel). Transgene and endogenous transcripts accumulated at similar levels. Gradual gain in transcript levels of key PrE TFs that are downstream targets of GATA6 is shown in the bottom panel highlighting transcript accumulation within 4 h of GATA6 expression. **b** GATA6 peaks associated with PrE genes identified by CUT&RUN were classified as proximal (within 5 kb of TSSs) or distal CREs and as late peaks (bound by GATA6 only at 48 h) and early peaks (bound by GATA6 by 8 h). Based on ATAC-seq signal, early peaks were further categorized into Closed at 0 h or Open at 0 h. Heatmap compares differentiation-induced changes at the different peak categories in GATA6 binding, corresponding ATAC-seq signal, as well as histone marks associated with active (H3K27ac, H3K4me3) and repressive (H3K9me3, H3K27me3) chromatin. **c** Browser view of the *Pdgfra* locus. Highlighted regions depict changes in accessibility, binding by PrE TFs (GATA6, GATA4, SOX17), and changes in active and repressive histone marks. Two *Pdgfra*-putative CREs are shown, one accessible and one closed at 0 h. **d** Median profile plots of normalized GATA6, ATAC, and H3K27ac signals at early GATA6 peaks closed at 0 h shows that GATA6 binding preceded increases in accessibility and H3K27ac. **e** Nucleosomal fraction of ATAC-seq signal as determined by ATACseqQC at the indicated timepoints over GATA6 motifs in two types of GATA6 early peaks: accessible before the onset of differentiation (left panel), and accessible only after GATA6 binding (right panel). In both peak types nucleosomal positioning over GATA6 motifs dropped quickly following GATA6 induction with stronger decrease at sites that were not accessible at 0 h. Source data are provided as a Source Data file.

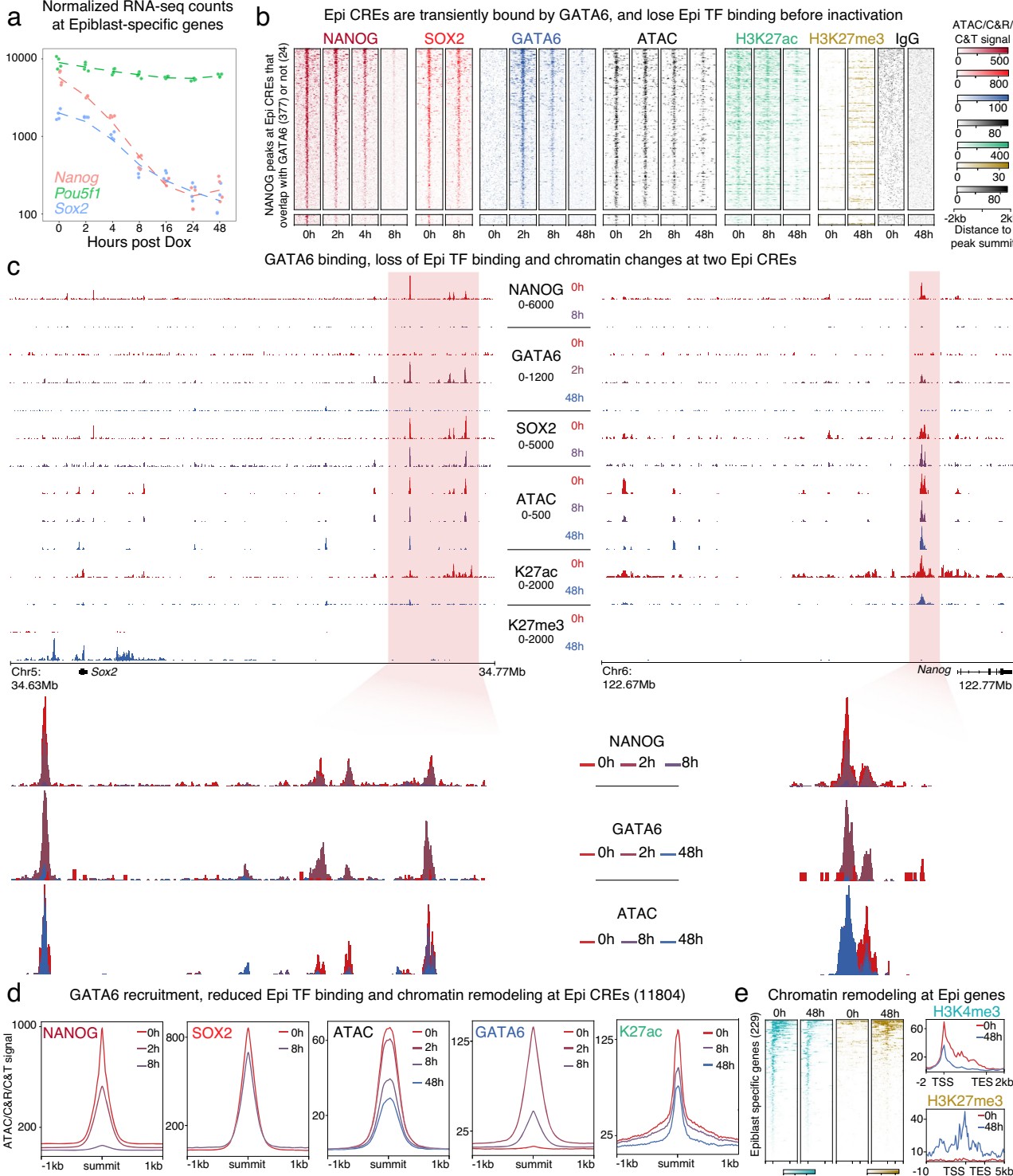

**Fig. 3 Inactivation of Epi CREs is preceded by transient GATA6 binding. a** Changes in normalized RNA-seq counts of three Epi genes during differentiation. As in blastocysts, *Nanog* mRNA levels decreased faster than *Sox2*, while *Pou5f1* was less affected **b** Heatmap shows gradual loss of NANOG and SOX2 at Epi CREs, concomitant with the transient binding of GATA6 at these loci, and progressive loss of ATAC signal and H3K27ac. A few CREs displayed increase in H3K27me3. **c** Browser view showing changes of NANOG, SOX2, and GATA6 binding, along with changes in ATAC signal, H3K4me3 and H3K27me3 at the *Nanog* and *Sox2* loci. While *Nanog* was silenced independently of H3K27me3, *Sox2* accumulated this mark over its gene bodies and surrounding regions. **d** Median profile plots depicting quick loss of NANOG and SOX2 with simultaneous GATA6 recruitment, and consequently a decrease in ATAC signal and H3K27ac. **e** Change in H3K4me3 and H3K27me3 surrounding Epi TSSs at 0 and 48 h. Some Epi genes accumulated H3K27me3 during differentiation.

At NANOG-bound regions, genome-wide and specifically at Epi CREs, we also detected strong SOX2 binding (Fig. 3b, c, Supplementary Fig. 3b). Strikingly, most of these sites were occupied by GATA6 starting as early as 2 h post Dox induction (Fig. 3b, top cluster, 377 peaks) and very few NANOG-SOX2 peaks were not bound by GATA6 (Fig. 3b, bottom cluster, 24 peaks). In line with its role in repression of *Nanog* and other Epi genes, GATA6 was previously shown to bind near the promoter of some pluripotency genes[35]. By profiling the time points immediately following GATA6 induction, we show here that GATA6 binding at Epi CREs is much more extensive than previously described and may impact the Epi program more profoundly. Contrary to PrE CREs, where GATA6 was bound throughout differentiation, Epi CREs and other NANOG-bound sites accumulated GATA6 transiently and only until 8 h (Fig. 3b–d–fourth panel). Concomitant with transient GATA6 binding, NANOG diminished progressively from 2 h onwards (Fig. 3b, d, first panel). SOX2 binding at Epi CREs diminished only slightly (Fig. 3b, d, second panel), likely because of its depletion at the protein level occurring only beyond 8 h (Supplementary Fig. 3a). NANOG was completely displaced from its target sites by 8 h, which coincided with the progressive loss of chromatin accessibility (Fig. 3b, black; Fig. 3d, third panel, Supplementary Fig. 3b) and reduction in H3K27ac levels (Fig. 3b, green; Fig. 3d, fifth panel, Supplementary Fig. 3b). A loss of H3K4me3 was seen at the promoters of Epi genes (Fig. 3e, blue) with a gain of H3K27me3 at some, but not all Epi promoters (Fig. 3e, yellow). In contrast to activation of PrE CREs, where nucleosome repositioning preceded CRE activation, nucleosome occupancy at NANOG motifs within its target CREs, remained unchanged despite the reduction in accessibility and H3K27ac levels (Supplementary Fig. 3c).

Because GATA6 binding precedes the chromatin landscape changes occurring at NANOG-bound CREs, we speculate that GATA6 induces eviction of the Epi TFs, either directly or indirectly, to achieve repression of the Epi transcriptional network. Transient recruitment of GATA6 was seen at both *Sox2* and *Nanog* CREs (Fig. 3c)[47,48]. These CREs showed a loss of NANOG-SOX2 binding, followed by reduction in chromatin accessibility, which trailed GATA6 binding at 2 h. Interestingly, *Sox2* accumulated H3K27me3, while *Nanog* only showed a decrease in H3K27ac with no accumulation of H3K27me3 (Fig. 3c). Since *Nanog* levels decreased faster than *Sox2* during PrE induction (Supplementary Fig. 3a), we propose that the repressive mechanisms employed by GATA6 at the two genes may reflect the difference in rate of transcriptional silencing. We did not characterize GATA4 and SOX17 binding at Epi CREs shortly after induction as these GATA6 targets are not yet expressed at the very early time points at which GATA6 transiently binds Epi CREs. Interestingly, GATA4 and SOX17 were not found at Epi CREs during later time-points of GATA6-induction (Supplementary Fig. 3f) suggesting that these factors aid GATA6 exclusively in activating the PrE program.

We wanted to further understand if GATA6 induces silencing of the Epi program by direct or indirect binding. Since NANOG- and SOX2-bound regions are highly accessible (Fig. 3b, Supplementary Fig. 3b), it is possible that GATA6 is attracted to these sites simply because of their high accessibility. Additionally, if the detected GATA6 peaks at NANOG-bound regions contained the GATA6 recognition motif, it would argue that GATA6 binding could be direct. Indeed, we found GATA6 motifs within CREs that were commonly bound by both NANOG and GATA6 at 2 h suggesting that inactivation of these regions is likely mediated by direct GATA6 recruitment (Supplementary Fig. 3d). Although Epi CREs contained motifs for both GATA6 and SOX2, they do not overlap frequently. To then address why NANOG-bound CREs recruited GATA6 less stably than PrE CREs, we compared the density of GATA6 binding motifs within the two types of GATA6 targets. Regions bound by NANOG and GATA6 contained lower density of GATA6 motifs as compared to GATA6 peaks not overlapping with NANOG at 0 h (Supplementary Fig. 3d, boxplot). These peaks, which we consider to be non-PrE GATA6 targets, also contained GATA6 motifs of weaker strength as measured by sequence similarity to the canonical GATA6 motif (Supplementary Fig. 3d, violin-plot). Analysis of CUT&RUN cut-site probability at GATA6 motifs located both in NANOG 0 h overlapping (Epi specific) and non-overlapping (PrE-specific) peaks provided further support that GATA6 does indeed occupy its motifs directly at both peak types (Supplementary Fig. 3e). The difference in GATA6 motif strength and density may explain why GATA6 binds Epi CREs only transiently while it binds PrE CREs more stably. It is also likely that GATA6 binding at Epi CREs is facilitated by NANOG occupying these regions and maintaining a highly accessible state. In line with this, GATA6 binding is reduced at 8 h, when NANOG levels decrease, together with lower accessibility.

### Evicted NANOG and SOX2 transiently bind GATA6-bound PrE CREs.
Despite being displaced from Epi CREs, NANOG and SOX2 protein levels remained stable for at least 4 h post Dox treatment (Supplementary Fig. 3a). Unexpectedly, we detected a transient doubling in the total number of NANOG and SOX2 peaks at 2 h compared to 0 h, which decreased back to initial numbers by 4 h (total peaks identified for NANOG: 0h-19995, 2h-41877, 4h-23519) (Supplementary Data 1). Since NANOG/SOX2-bound CREs were capable of recruiting GATA6, we wondered if the NANOG/SOX2 peaks gained at 2 h were associated with GATA6-bound CREs. Strikingly, a large number of 2h-specific NANOG peaks appeared at GATA6-bound CREs made accessible after GATA6 binding at 2 h (Fig. 4a, blue cluster). Interestingly, while peaks that were bound by NANOG before GATA6 induction began, progressively lost NANOG (Fig. 4a, red cluster), sites made accessible by GATA6 (blue cluster) showed increased NANOG binding (Fig. 4b–d). Importantly, GATA6-bound PrE CREs also accumulated NANOG and SOX2 (Supplementary Fig. 4a). These observations suggest that NANOG and SOX2 evicted from Epi sites were redirected to GATA6-bound CREs. To validate this observation with a different technique, we performed chromatin immunoprecipitation coupled with sequencing (ChIP-seq) for FLAG (to map GATA6 binding) and NANOG at 0 and 2 h. In agreement, with our observations from CUT&RUN, the transient increase in NANOG binding at GATA6-bound CREs, specifically at 2 h, was also evident by ChIP-seq (Supplementary Fig. 4b).

We next considered the possibility that NANOG and SOX2 were redirected to GATA6-bound regions at 2 h because these sites contain recognition motifs for the pluripotency factors. To address this, we focused on GATA6 target sites to look for enrichment of the murine OCT4/SOX2 consensus motif, which is better defined than the NANOG motif. We found that GATA6 peaks indeed contained the OCT4/SOX2 motif, although at lower density (Fig. 4e, boxplot) and of weaker motif strength (Fig. 4e, violin-plot) than the motifs within pluripotency CREs (peaks bound by NANOG-SOX2 at 0 h). The presence of binding motifs for Epi TFs, within GATA6-bound CREs, suggests that the PrE transcriptional network can be directly regulated by pluripotency TFs. Together with Fig. 3, these data suggest that GATA6 and NANOG may regulate common CREs which could allow them to control lineage specification into either the PrE or Epi in blastocysts.

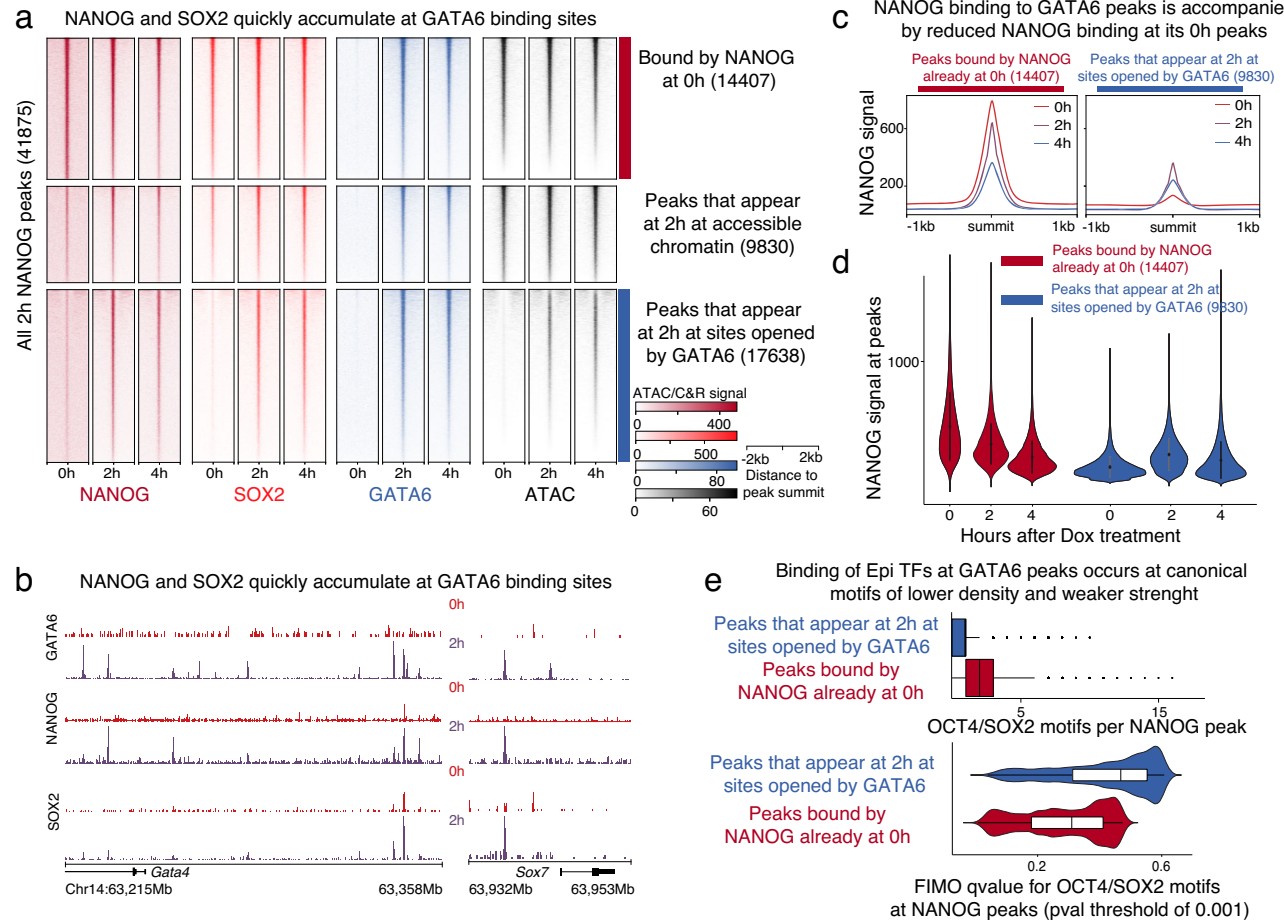

**Fig. 4 Evicted pluripotency factors transiently occupy GATA6-bound PrE regions. a** Heatmap showing NANOG, SOX2, and GATA6 binding, together with ATAC signal at regions of the genome bound by NANOG at 2 h. These NANOG peaks were divided in three types: occupied already at 0 h (cluster 1-red), bound de novo by NANOG at 2 h and accessible at 0 h (cluster 2), and bound de novo by NANOG at sites closed at 0 h (cluster 3 -blue). **b** Representative browser view showing the de novo recruitment of NANOG and SOX2 at 2 h at GATA6 peaks found at putative CREs controlling PrE genes GATA4 and SOX7. **c** Median profile plots showing a reduction in NANOG binding at NANOG 0 h peaks (left panel, cluster 1 from A) with a simultaneous increase in NANOG binding at peaks opened by GATA6 binding (right panel, cluster 3 from A). **d** NANOG CUT&RUN signal from 0, 2, and 4 h post Dox induction plotted at peaks comprising clusters 1 and 3 in 4a. The violin plot shows a decrease of read density at cluster 1 peaks with a simultaneous increase at cluster 3 peaks. **e** Peak per motif density and strength of OCT4-SOX2 binding motifs within NANOG peaks that appear at 2 h (blue) compared to NANOG peaks defined at 0 h (red). Motif strength is represented by the *q*-value of the FIMO analysis comparing the motif found at peaks to the consensus sequence. Boxplots show minimum, maximum, median, first, and third quartiles. Source data are provided as a Source Data file.

**GATA6 and NANOG co-occupy Epi and PrE CREs also in vivo.** Although our CUT&RUN data indicates that GATA6 and NANOG co-localize at EPI and PrE CREs, this assay is not able to distinguish if GATA6 and NANOG co-bind the same nucleosomes. To determine if GATA6 and NANOG indeed co-bound on the same CREs we performed sequential ChIP (reChIP). We first immunoprecipitated chromatin at 0 h and 2 h with a FLAG antibody to identify GATA6-bound CREs. Then, we reprecipitated the eluted FLAG-bound chromatin with a NANOG antibody. We plotted the signal from the GATA6-FLAG ChIP, and NANOG reChIP, at the subset of Epi and PrE CREs, which we defined in Fig. 1d and that showed progressive loss and gain in accessibility, respectively. In line with Figs. 2 and 3, FLAG-signal was enriched at both Epi and PrE CREs suggesting they are bound by GATA6. Even though NANOG binds Epi CREs at high frequency in undifferentiated cells, we did not obtain any specific enrichment of NANOG reChIP signal at 0 h. Since GATA6 expression and binding is absent at this time point this indicates that we did not have carryover of nonspecific IP material in our reChIP. In contrast, at 2 h we saw a strong enrichment of

NANOG reChIP signal at GATA6 bound Epi and PrE CREs, supporting that the two TFs bound together at these sites (Fig. 5a). An example of co-binding is evident at CREs proximal to *Pdgfra* (PrE gene, Fig. 5b, left panel) and *Klf4* (Epi gene, Fig. 5b, right panel), as well as at GATA6-bound locations genome-wide (Supplementary Fig. 5a).

We then asked if NANOG and GATA6 bound the same Epi and PrE CREs in unspecified ICM cells in vivo. We used CUT&RUN to profile NANOG and GATA6 binding in early (E3.25-3.5) blastocysts (staged as in Supplementary Fig. 5b), a stage when the two TFs are co-expressed. Because specification of the PrE and Epi fates is a rapid process, not occurring at the same time and pace in all ICM cells, we suspected that in blastocysts, NANOG and GATA6 binding at their target sites could be very transient. To robustly recover NANOG and GATA6 targets in blastocysts, we modified the CUT&RUN protocol to include light chromatin fixation. Additionally, instead of bead-assisted binding, we adapted methodology used for immunofluorescence in blastocysts[4,49], which involves manual handling of embryos under the microscope at each step of the protocol and between washes.

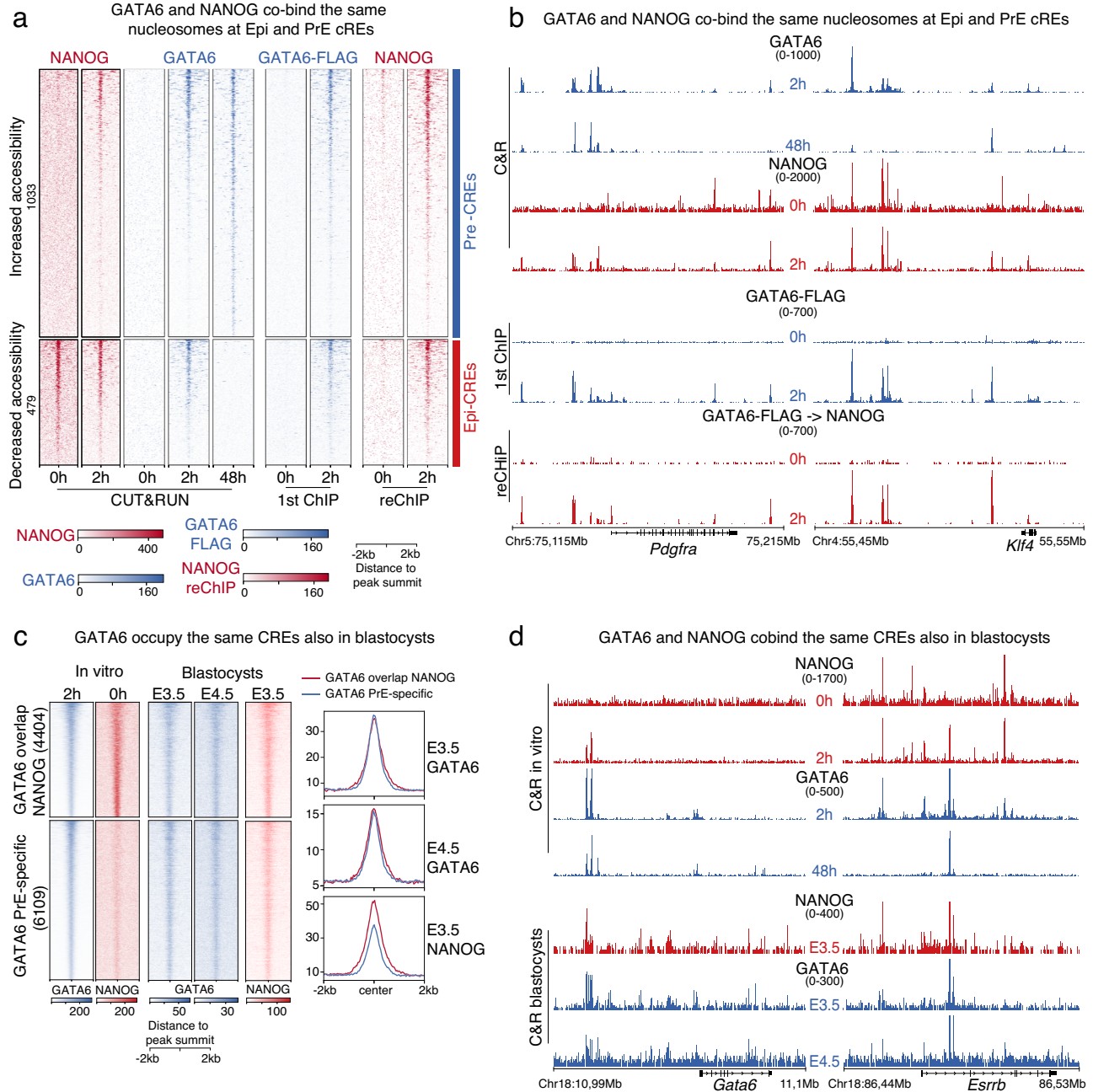

**Fig. 5 GATA6 and NANOG co-bind at Epi and PrE CREs in vitro and in vivo. a** Heatmap showing enrichment of NANOG reChIP signal at FLAG-bound Epi and PrE CREs that lose and gain accessibility during PrE differentiation. **b** Representative browser view showing that at 2 h NANOG and GATA6 co-bind at CREs proximal to the *Pdgfra* and *Klf4* genes. GATA6 and NANOG CUT&RUN signals are included to compare with signal from reChIP. **c** Extensive co-binding of GATA6 and NANOG to Epi-CREs also occurs in blastocysts. Heatmap and median profile plot comparing GATA6 and NANOG CUT&RUN patterns in blastocysts, to in vitro GATA6 and NANOG binding at 0 h and 2 h of GATA6-driven differentiation. **d** Example of GATA6 binding both at Epi (*Esrrb*) and PrE loci (*Gata6*) in blastocysts. Source data are provided as a Source Data file.

Our modified protocol prevented loss of biological material, preserved antibody recognition and physical attributes of the embryo (Supplementary Fig. 5b), and significantly improved our ability to recover high quality genome-wide TF binding data. CUT&RUN data obtained in vivo was plotted at regions bound by NANOG at 0 h in vitro, where GATA6 transiently bound at 2 hrs (Fig. 5c, top cluster) and regions bound by GATA6 as early as two hours but not occupied by NANOG in undifferentiated cells (Fig. 5c, bottom cluster). The top cluster includes Epi CREs while the bottom cluster represents PrE CREs. In early blastocysts, both NANOG and GATA6 binding was detected at

regions bound by NANOG at 0 h in vitro (Fig. 5c, top cluster). Interestingly, both TFs were also found bound at GATA6-specific peaks (Fig. 5c, bottom cluster). Importantly, both GATA6 peak types exhibited similar GATA6 binding signal intensity (Fig. 5c, top right plot). Expectedly, NANOG was more enriched at GATA6-NANOG shared peaks, as compared to GATA6-specific PrE peaks (Fig. 5b, bottom right plot). A good example of GATA6 and NANOG binding the same CREs is evident at both the *Gata6* and *Esrrb* loci (Fig. 5d). These data support the idea that GATA6 can occupy the same targets as NANOG in developing embryos. We also found GATA6 binding in late blastocysts, isolated at

E4.5, when PrE fate determination is completed. Surprisingly, like in early blastocysts, late-blastocysts exhibited comparable preference of GATA6 binding at NANOG-GATA6 shared sites, as well as GATA6-specific PrE-sites (Fig. 5c), which suggests that even in late blastocysts, GATA6 continues to bind Epi CREs in PrE committed cells, where its occupancy may serve to repress these elements. Together, our assays demonstrate that the binding of NANOG and GATA6 to both Epi and PrE CREs occurs in vitro and in blastocysts. This extensive shared binding at CREs controlling the Epi and PrE transcriptional networks may facilitate rapid bifurcation of precursor cells into divergent lineages.

**Extensive rewiring of enhancer contacts upon GATA6 induction.** As PrE induction is associated with TF binding redistribution and changes in chromatin landscape, we next investigated if these changes affected global three-dimensional (3D) genome organization. Chromatin conformation assays have shown that large chromosomal domains, several megabases in length, can reside in either the active (A) or repressive (B) nuclear compartments depending on their transcriptional state and the histone modifications they harbor[50]. Because PrE induction was associated with changes in H3K27ac across the genome (Figs. 2b and 3b), we first used Hi-C to profile 3D-genome organization of uninduced (0 h) and PrE-like (48 h) cells. Despite changes in chromatin landscape, genome organization at the compartment level was markedly similar in undifferentiated and differentiated cells. PC1 eigenvalue scores from a PCA allowed identification of A and B compartment composition at 250 kb bins. As an example, organization of compartments on chromosome 1 is depicted in Fig. 6a, which shows how it remained largely unchanged even after 48 h of GATA6 expression. Just 1.4 percent of the entire genome showed differences in compartmentalization upon PrE-like differentiation (Fig. 6b, bottom cluster). These genomic regions showed changes in H3K27ac levels and included PrE-specific genes like *Sox17*, *Fgfr2* and *Lrp2* which gained H3K27ac and were reorganized from inactive to active compartment. On the other hand, Epi genes such as *Lef1*, *Chd9*, and *Kat2b* lost H3K27ac and moved from the A to B compartment. Interestingly, genes such as *Nanog, Sox2, Gata6* and *Gata4*, that showed dramatic changes in transcript levels, did not change compartments.

Besides large-scale chromatin reorganization, gene transcription can impact and be impacted by local interaction changes involving proximal and distal regulatory elements. TFs have been shown to influence genome topology to drive differentiation-associated expression changes[51–54]. While some reports have shown that remodeling of genome topology can be linked to transcriptional changes[55–59], contrasting studies at developmentally regulated genes[60–64] have shown that transcriptional changes can occur independently of chromatin structure reorganization. These opposing results suggest that the contribution of 3D-genome interactions is locus and context specific. Since PrE specification involves rapid chromatin remodeling, we wondered if this short time frame was sufficient to alter genome topology at key PrE and Epi genes. To address this, we performed Capture-C to assess changes in interactions between CREs controlling Epi and PrE genes that showed the highest transcriptional changes between uninduced (0 h) and PrE-like cells (48 h). At all PrE-specific genes tested, with the exception of *Fgf3*, we detected a gain in interaction between the promoter and putative CREs marked by gain of H3K27ac (Fig. 6c, Supplementary Fig. 6) within 16 h of GATA6 expression. Contrarily, most Epi-genes assayed (except for *Morc1*, *Pou5f1*, and *Fgf4*), were associated with a reduction of interactions with their CREs, which

lost H3K27ac during PrE induction (Fig. 6d, Supplementary Fig. 7). Interestingly, we could not identify distal lineage-specific CREs for any of the four genes without changes (*Fgf3*, *Fgf4*, *Morc1*, *Pou5f1*) in interactions during GATA6 induction. As these genes are likely regulated only by proximal CREs, this might explain the stability of 3D interactions upon GATA6 induction. Genes not expressed in either lineage were used as controls and as expected, showed no change in interactions from their promoter regions (Supplementary Fig. 7, *Cdx2* and *Pax5*). In addition to increase in H3K27ac, all PrE CREs tested were associated with GATA6 binding in their vicinity. Likewise, Epi CREs where interactions decreased, showed a loss of NANOG and SOX2 occupancy initiated by GATA6 binding at these sites. Thus, even though induction of a PrE fate is associated with limited large-scale genome reorganization, we observed significant changes in interactions between CREs controlling genes forming the Epi and PrE networks. Together our data show how GATA6, the PrE-specifying TF, initiates a cascade of interdependent mechanisms to regulate the genome to specify the PrE lineage and inhibit the alternative Epi fate.

## Discussion

Multicellular organisms comprise of diverse cell-types which often arise from a common progenitor. scRNAseq studies have highlighted how this is accomplished by combinatorial TF modules that promote multifurcation of cell-types from common multipotent progenitors[2]. Using GATA6-driven transdifferentiation of ES cells, we propose how NANOG and GATA6 maintain plasticity in bipotent ICM cells, and how higher GATA6 levels preferentially promote a PrE fate. ES cells between 2 to 8 h of differentiation co-express GATA6 and NANOG, and their transcriptome resembles the ICM in E3.25 blastocysts or uncommitted progenitor cells in later blastocysts. During co-expression (at 2 h) we detected GATA6 co-binding along with NANOG at both Epi and PrE CREs. This was corroborated by our observations in blastocysts, where we also detected NANOG and GATA6 binding at both Epi and PrE CREs (Fig. 5). In uncommitted ICM cells, where NANOG and GATA6 levels may be equivalent, we propose that both factors are bound at their recognition motifs within Epi and PrE CREs, maintaining ICM cells in a poised state to adopt either cell fate.

When higher GATA6 levels are reached, relative to NANOG (4 h in vitro), we observed that pluripotency TFs are evicted from Epi CREs and redirected to PrE CREs, which we propose may aid GATA6 in achieving quick activation of the PrE network. Similar TF mobilization has been described during reprogramming of fibroblasts into induced pluripotency stem cells. The reprogramming TFs–OCT4, SOX2 and KLF4 (OSK)–were shown to inactivate the somatic-cell transcriptional network by disengaging somatic TFs from their target sites and redistributing them to pluripotency loci engaged by OSK[65] to facilitate activation of the pluripotency program. Similarly, the transition from naïve to primed pluripotency in Epi cells of the implanted embryo is facilitated by redistribution of OCT4[66]. In this case, OCT4 relocalization from naïve- to primed- pluripotency genes, was initiated by the primed-pluripotency specific TF, OTX2. OCT4 redistribution was preferentially seen at sites carrying abundant and strong OTX2 recognition motifs. We also propose that the inherent differences in the strength and density of GATA6 binding sites in PrE- compared to Epi CREs may enable a shift in binding patterns of pluripotency factors when GATA6 levels are high, thus promoting an exit from plasticity and differentiation into PrE.

Supporting this hypothesis we found that PrE CREs contain higher density of high-affinity GATA6 recognition motifs

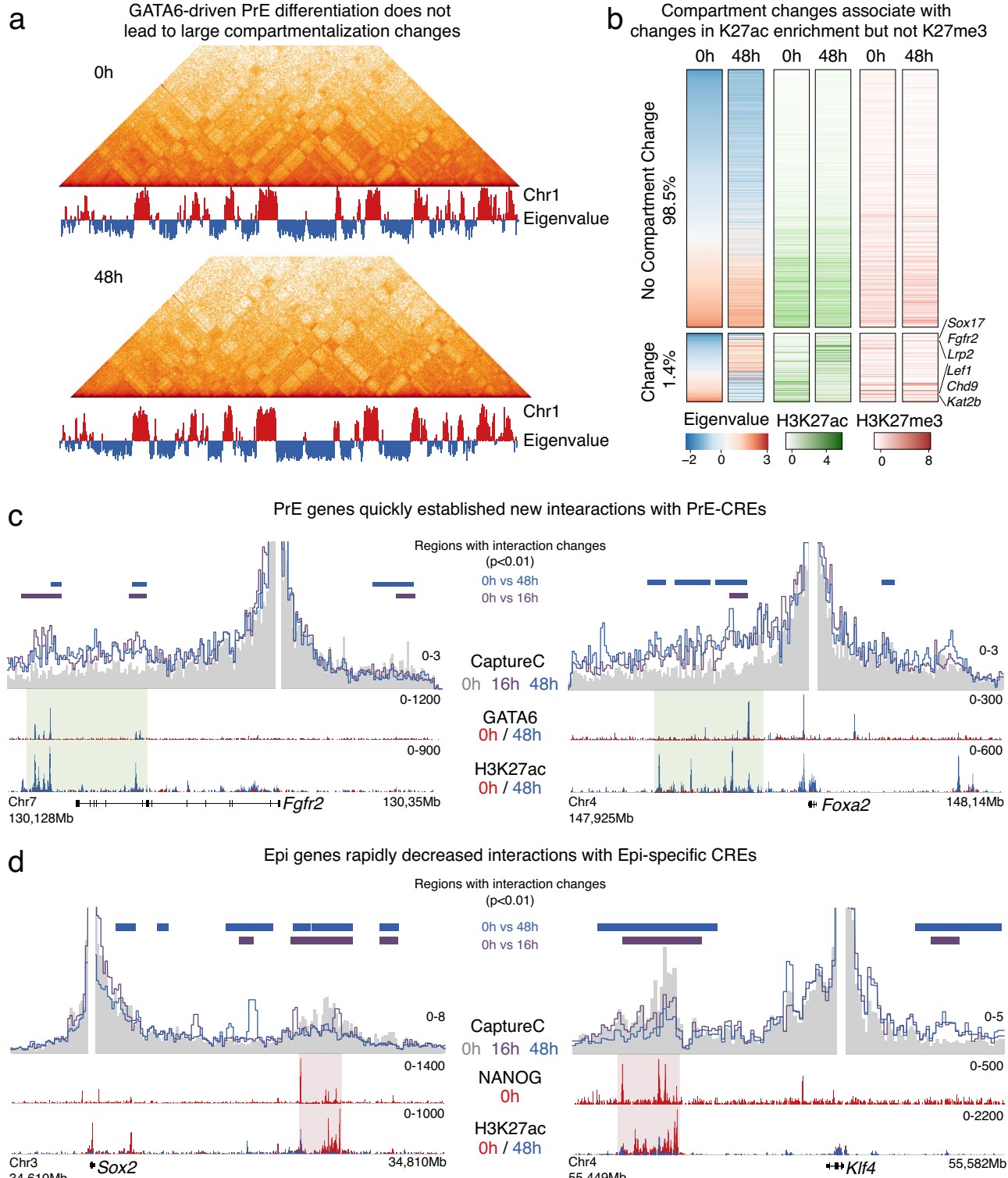

**a** GATA6-driven PrE differentiation does not lead to large compartmentalization changes

**b** Compartment changes associate with changes in K27ac enrichment but not K27me3

**c** PrE genes quickly established new interactions with PrE-CREs

**d** Epi genes rapidly decreased interactions with Epi-specific CREs

compared to Epi CREs. This difference in motif affinity and density could provide a distinguishing feature enabling GATA6 to achieve contrasting changes at PrE and Epi CREs. A scenario can be envisioned where CREs carrying higher densities of strong TF motifs would be bound longer and more stably by the TF, marking them for transcriptional activation. Contrarily, less abundant, lower affinity binding sites could lead to only transient occupancy by the TF, not allowing sufficient time for activation, and instead marking them for inactivation by recruitment of histone deacetylases. A related observation was recently made

where cells of the caudal epithelium that contain neural and mesodermal progenitors use binding of the same TF (SOX2) at sites of different affinity to elicit opposing regulatory responses in cells differentiating into divergent paths[67]. In our study, we observed that CREs activated during PrE differentiation were bound by GATA6 starting at 2hrs and continued in most cases until 48 h (Fig. 2b). CREs inactivated by GATA6, on the other hand, were bound by GATA6 only transiently at 2 h (Fig. 3b), supporting that GATA6 residence time at its targets could affect transcriptional activity. This, however, remains to be tested

**Fig. 6 GATA6-induced chromatin changes were followed by fast rewiring of interactions between Epi and PrE-specific genes with their CREs. a** Intra-chromosomal interactions over chromosome 1 detected by Hi-C and first principal component eigenvalue show that nuclear compartments composition was mostly unchanged during differentiation. **b** Heatmap showing eigenvalues of first principal component of Hi-C performed at 0 and 48 h. Less than 2% of all 250kb-bins changed their compartment during PrE differentiation (bottom cluster). Most 250 kb-bins did not change compartment (top cluster). Clusters are not represented to scale. Changes in compartments were associated with changes in H3K27ac but no changes were observed for H3K27me3 levels. Bins that changed from compartment B to A gained K27ac signal and bins that changed from B to A, lost this histone modification. Bins were sorted based on eigenvalue at 0 h. Examples of Epi and PrE-specific genes whose compartment localization changed during differentiation are shown. **c, d** In contrast to stable compartment status, interactions between gene promoters and their putative CREs were quickly remodeled within 16 h of GATA6 expression. PrE genes shown in c, quickly increased interactions with sites bound by GATA6 that gained H3K27ac during differentiation. Epi genes in contrast, reduced interactions with CRES that were occupied by NANOG at 0 h and that lost H3K27ac following GATA6 expression. Capture-C data is shown as the average signal of two replicates using bins of 1 kb. Green shaded area represents putative CREs showing increased interactions with PrE-specific genes while red shaded areas highlight Epi-CREs that lose interactions with putative Epi-specific genes. $P$ values were calculated using the Wald test in DESeq2 and comparing Capture-C signal of 2 replicates over overlapping 5 kb windows across the regions shown in this figure. Comparisons were done between 0 h and either 16 h or 48 h. Adjusted $p$ values using the FDR/Benjamini-Hochberg option in DE-Seq2 that were lower than 0.01 were considered as significant. Horizontal bars represent windows considered as statistically significantly different for these comparisons.

thoroughly in future experiments where the presence and frequency of these motifs will need to be manipulated. Combinatorial binding by other lineage-restricted transcription factors could also contribute to marking loci for activation and to allow for input signal integration by different signaling pathways. This has been studied extensively and described in lineage specification of multiple cell types[68–72]. CREs controlling the PrE transcriptional network carry high affinity binding sites for GATA6 (Supplementary Fig. 3d) and are also bound by its downstream targets SOX17 and GATA4 (Supplementary Fig. 2d) potentially functioning to mark PrE genes for robust activation rather than repression.

In addition to NANOG and GATA6 levels, FGF/ERK signaling via FGF4 is essential for lineage fate choice in the ICM. In $Fgf4^{-/-}$ blastocysts, all ICM cells adopt an Epi fate at implantation, even though GATA6-expressing PrE-precursors are initially present, showing that PrE fate determination is dependent on FGF4/ERK signaling[5,17–20,73–77]. A mechanism where FGF4 regulates the ability of ES cells to exit pluripotency and differentiate into PrE was described recently[78]. Using an in vitro system where levels of ERK activation can be modulated, Hamilton et al. showed that enhancer activity was proportionally affected merely by reversible dissociation of cofactors and the transcriptional machinery without TF redistribution. This provides an explanation for how stochastic fluctuations in FGF4 availability can be incorporated at the transcriptional level to promote divergent cell-fate determination in ICM cells. It is noteworthy that the extent of FGF/ERK activation used in this study, while sufficient to affect the degree of pluripotency was not enough to upregulate GATA6. This suggests that the observed changes in FGF signaling, recapitulated stochastic initiation of fate determination in early blastocysts where skewed ratios of GATA6 and NANOG are not yet established. Our study on the other hand, describes events starting from co-expression to establishment of skewed GATA6:NANOG ratio and hence recapitulates the process by which PrE fate-determination is initiated and completed.

While our study describes an unanticipated behavior of GATA6 and NANOG during a stage of co-expression, there are a few caveats associated with our approach. In blastocysts, PrE cells directly differentiate from cells of the ICM. In contrast, ES cells in vitro resemble Epi cells in blastocysts more closely, and GATA6 induction leads to establishment of a PrE state through a transdifferentiation process that forms intermediate stages that transcriptionally resemble the ICM. Because the trajectory of in vitro differentiation differs from that of blastocysts, it is possible that there are differences in molecular mechanisms occurring in vivo and in vitro, which we are unable to define with our current approach. To address this, we confirmed some of our

observations by performing CUT&RUN in pooled blastocysts (Fig. 5). However, at E3.5, blastocysts are composed of a mixture of ICM cells that co-express GATA6 and NANOG, and cells that have already committed to exclusively expressing only one of these TFs. Therefore, it is not possible to discriminate whether our data represent TF co-binding in vivo or TF-binding at the same CREs in different cells specified into divergent fates. This can only be effectively addressed by single-cell approaches, which are not yet optimal to study TFs in preimplantation embryos. In summary, we show that GATA6 and NANOG bind shared regulatory elements in blastocysts, however whether this occurs simultaneously, and the precise function of such shared binding, remains to be tested thoroughly.

The extensive TF redistribution evident during PrE specification is accompanied by rapid local chromatin reorganization. GATA6 binding at PrE and Epi CREs initiates a cascade of changes at multiple layers of chromatin. By altering the levels of H3K27ac, GATA6 regulates enhancer activity, which coincides with rewiring of genome structure. H3K27ac levels have been shown to influence enhancer-promoter interactions[57,58], however it is unclear if enhancer activity results in rewiring of interactions, or vice versa. It is plausible that in unspecified ICM cells, fluctuations in enhancer activity are first initiated and once GATA6 is expressed above a certain threshold compared to NANOG, PrE enhancers are stably activated (and Epi enhancers inactivated) leading to rewiring of enhancer-promoter interactions to facilitate continued enhancer activity. A temporal resolution of the two mechanisms, changes in histone modification and chromatin rewiring, would allow for the stepwise generation of a feedforward loop driving successive and unidirectional lineage commitment. Alternatively, it is also possible that TF-mediated multilayered chromatin regulation provides redundancy to ensure robust gene regulation. Further studies will be required to address these possibilities, and the blastocyst, owing to its simplicity and temporally distinct stages, provides an excellent self-contained system to gain mechanistic insights in vivo.

## Methods

**Cell line, culture conditions, in vitro differentiation, and animal experiments.** Previously generated mouse embryonic stem cells (mES) expressing a single copy of a transgene containing a bidirectional TET-responsive element driving expression of doxycycline-inducible GATA6-FLAG and DsRed2 were a kind gift from Dr. Niakan's laboratory[35]. Cells were maintained at a density of 1 million cells per 10 cm gelatinized plates at 37 °C and 5% $CO_2$ in 2i media (50% DMEM/F12, 50% Neurobasal, Penicillin-Streptomycin, GlutaMAX, 2-Mercaptoethanol, N2, B27, 0.3 nM PD0325901, 0.1 nM CHIR9902, and $2 \times 10^5$ units/mL LIF). Differentiation into primitive endoderm was induced in serum containing ES media (Knockout DMEM, 15% FBS,1% GlutaMAX, 1% Penicillin-Streptomycin, 1% 2-Mercaptoethanol, 1% Sodium Pyruvate, 1% MEM NEAA, $2 \times 10^5$ units/mL LIF) supplemented with doxycycline (1 μg/mL) for 2 to 12 h. For time points greater than 12 h,

mES cells were induced in serum media containing doxycycline for 12 h and switched to serum media in the absence of doxycycline for the remaining time. All mouse experiments were performed according to NIH and PHS guidelines and only after protocols were approved by the Animal Care and Use Committees of the National Cancer Institute and Eunice Kennedy Shriver National Institute of Child Health and Human Development.

**Fluorescence assisted cell sorting**. Cells were harvested using 10 mM EDTA made in PBS. After incubation in EDTA for 10 mins, cells were dislodged by pipetting to ensure a suspension of single cells, following which cells were spun and resuspended in 500 μl of MACS buffer (PBS, 2% FBS, 1 mM EDTA). To analyze proportions of differentiating populations, cells were double stained with pre-conjugated PDGFRA-FITC (ThermoFisher Scientific, 11-1401-82) and PECAM-APC (BD Pharmigen, clone MEC 13.3, 551262) antibodies by adding 0.5 μl of each antibody. Cells were incubated at 4 °C for 20 mins following which they were washed twice in MACS buffer. Cell pellets were finally resuspended in 800 μl MACS buffer and analyzed for proportions of FITC positive/APC negative cells. To sort live dsRed2 positive populations for downstream analysis by RNAseq and ATAC-seq, cells were harvested by trypsinization and directly resuspended in MACS buffer. Cells were gated using uninduced, dsRed2 negative cells, as control. dsRed2 positive populations were collected in HBSS.

**Immunofluorescence**. Cells were seeded on 6-well dishes in 2i, at a density of 25,000 cells/well. After Dox induction, cells were rinsed twice in PBS and fixed using 4% formaldehyde for 10 mins, following which cells were washed thrice in PBS and then permeabilize for 10 mins at room temperature using 0.5% Triton X-100 made in PBS (0.5% PBST) supplemented with 100 mM glycine. Cells were again washed thrice in PBS and incubated in 2% horse serum, in a humidified chamber at 37 °C for 30 mins. Cells were then incubated in primary antibodies (SOX17-R&D Systems, AF1924, SOX2-Millipore, ab5603, GATA6- R&D Systems, AF1700, NANOG- Active Motif, 61419) diluted at 1:500 in 2% horse serum, 0.5% Triton-X-100 made in PBS, at 4 °C overnight. Cells were then washed thrice in 0.1% PBST and incubated in secondary antibodies (prepared in PBS at 1:1000; Donkey-anti-rabbit-AF488, Invitrogen A21206; Donkey-anti-goat-AF555, Invitrogen A21432) for 1 h at room temperature, protected from light. Secondary antibody was then replaced with Hoechst solution (10 μg/ml in PBS) for 2 mins and then cells were thoroughly washed in 0.1% PBST before imaging at 10X on an inverted epifluorescence microscope.

**Western blot**. Whole cell extracts were prepared in 1X Radioimmunoassay buffer (RIPA, 50 mM Tris-Cl pH 8,150 mM NaCl, 2 mM EDTA pH 8, 1% NP40, 0.5% sodium deoxycholate, 0.1% SDS) supplemented with protease inhibitor cocktail (Roche). Protein content was estimated using the Peirce BCA kit, and equal amounts of protein (20μg) resolved on 10% SDS-PAGE gels. Proteins were transferred onto methanol-activated PVDF membrane (Immobilon-FL), followed by blocking (5% milk made in TBST) and incubation with primary antibodies. Proteins were detected using HRP-conjugated secondary antibodies (Cell signaling technologies-7074P2). Primary antibodies (and dilutions) are: Rabbit anti-NANOG (ab80892; 1:1000), Rabbit anti-SOX2 (ab92494; 1:1000), histone-H3 (ab176842; 1;1000). Uncropped images of western blot, together with molecular weight markers, can be found at the end of Supplementary Figures.

**RNA extraction, RNA-seq library preparation and sequencing**. RNA from four replicates at each time point was isolated using trizol. After confirming that the RNA integration number for each sample was above 8, libraries were prepared using TruSeq Stranded mRNA prep kit with PolyA purificaton and sequenced on HiSeq 2500 using SE50.

**ATAC-seq library preparation and sequencing**. ATAC-seq was performed, in duplicates, as described in[79]. Briefly, cells at each time point were dissociated using Accutase, and 50,000 cells were subjected to the tagmentation reaction. Cells were first washed in resuspension buffer (10 mM Tris-HCl pH 8.0, 10 mM NaCl, and 3 mM MgCl$_2$ in water), following which nuclei were isolated in 1 ml lysis buffer (10 mM Tris-HCl pH 8.0, 10 mM NaCl, 3 mM MgCl$_2$, 0.1% NP-40, 0.1% Tween-20, and 0.01% Digitonin in water) on ice for three min. Nuclei were rinsed once in wash buffer (10 mM Tris-HCl pH 8.0, 10 mM NaCl, 3 mM MgCl2, and 0.1% Tween-20) and tagmentation was carried out using 2.5 ul Tn5 transposase (Illumina 15027865) for exactly 30 mins. Following tagmentation, DNA was purified using the Zymo DNA Clean and Concentrator kit (Zymo, D4033). Libraries were prepared using Q5 polymerase and unique indices were added to each sample. First, gap filling was performed at 72 °C for 5 mins followed by five cycles of 98 °C, 20s, 63 °C, 30s, and 72 °C 1 min. After initial amplification, tubes were held on ice, while quantitative PCR was run on 1 μl of the pre-amplified library to determine additional number of cycles needed. Libraries were sequenced on HiSeq2500 using PE50.

**CUT&RUN**. CUT&RUN was performed, in duplicates, as previously described[80] with small modifications. Briefly, mES cells were dissociated using Accutase,

counted, and 100,000 cells/replicate were pelleted at 600 g for 3 min at room temperature. Supernatant was discarded, cells were resuspended in Wash Buffer (20 mM HEPES pH 7.5, 150 mM NaCl, 0.5 mM Spermidine, 1x Protease inhibitor cocktail), and pelleted at 600 g for 3 min at room temperature. BioMag Plus Concanavalin A beads (Bangs Laboratories) were equilibrated in Binding Buffer (20 mM HEPES pH 7.5, 10 mM KCl, 1 mM CaCl$_2$, 1 mM MnCl$_2$). Cells were resuspended in Wash Buffer, mixed with a slurry of equilibrated Concavalin A coated magnetic beads, and rotated for 10 min at room temperature. Per 100,000 cells, 10 μl bead slurry was used. Beads were placed on a magnetic separator and supernatant was discarded. Beads were resuspended in Wash Buffer containing 2 mM EDTA, 0.1% bovine serum albumin, 0.05% Digitonin, and 1:50 dilution of primary antibody. This was incubated on a nutating platform for 2 h at room temperature. After incubation, beads were washed twice in Digitonin Buffer (20 mM HEPES pH 7.5, 150 mM NaCl, 0.5 mM Spermidine, 1x Roche Complete Protease Inhibitor no EDTA, 0.05% Digitonin and 0.1% bovine serum albumin), then incubated with pA-MNase (600 μg/ml, 1:200, either home-made or a gift from Steven Henikoff) in Digitonin Buffer for 1 h at 4 °C. After incubation, beads were washed twice, resuspended in 150 μl of Digitonin Buffer, and equilibrated to 0 °C before adding CaCl$_2$ (2 mM) and incubating for 1 h at 0 °C. After incubation, 150 μl of 2X Stop Buffer (200 mM NaCl, 20 mM EDTA, 4 mM EGTA, 50 μg/ml RNase A, 40 μg/ml glycogen), was added. Beads were incubated for 30 min at 37 °C and then pelleted at 16,000 g for 5 min at 4 °C. Supernatant was transferred, mixed with 3 μL 10% SDS and 1.8U Proteinase K (NEB), and incubated for 1 h or overnight at 55 °C, shaking at 900 rpm. After incubation, 300 μl of 25:24:1 Phenol/Chloroform/Isoamyl Alcohol was added, solutions were vortexed, and transferred to Maxtrack phase-lock tubes (Qiagen). Tubes were centrifuged at 16,000 g for 3 min at room temperature. 300 μl of Chloroform was added, solutions were mixed by inversion, and centrifuged at 16,000 g for 3 min at room temperature. Aqueous layers were transferred to new tubes and DNA isolated through Ethanol precipitation and resuspended in 10 mM Tris-HCl pH 8.0 (ThermoFisher). CUT&RUN libraries were prepared following the SMARTer ThruPlex TAKARA Library Prep kit with small modifications. For each sample, double stranded DNA (10 μl), Template Preparation D Buffer (2 μl), and Template Preparation D Enzyme (1 μl) were combined, and End Repair and A-tailing was performed in a Thermocycler with a heated lid (22 °C, 25 min; 55 °C, 20 min). Library Synthesis D Buffer (1 μl) and Library Synthesis D Enzyme (1 μl) were subsequently added, and library synthesis was performed (22 °C, 40 min). Immediately after, Library Amplification D Buffer (25 μl), Library Amplification D Enzyme (1 μl), Nuclease-free water (4 μl), and a unique Illumina-compatible indexed primer (5 μl) were added. Library amplification was performed using the following cycling conditions: 72 °C for 3 min; 85 °C for 2 min; 98 °C for 2 min (denaturation); 4 cycles of 98 °C for 20 s, 67 °C for 20 s, 72 °C for 10 s (addition of indexes); 14 cycles of 98 °C for 20 s, 72 °C for 10 s (library amplification). Post-PCR clean-up was performed on amplified libraries with a SPRIselect bead 0.6X left/1x right double size selection then washed twice gently in 80% ethanol and eluted in 10–12 μl 10 mM Tris pH 8.0. 1:50 dilutions of primary antibodies against the following TFs were used: GATA6 (R&D Systems, AF1700), NANOG (Active Motif, 61419), SOX2 (Millipore, ab5603), GATA4 (Santa Cruz Biotech, sc25310), SOX17 (R&D Systems, AF1924), H3K9me3, (Abcam, ab8898).

**CUT&RUN in blastocysts**. Blastocysts were collected from 4–5 weeks old C57BL/6 N females six days after PMSG/HCG injections. Blastocysts were used right away to profile E3.5 embryos, after removing the zona pellucida using acid tyrode's solution. E4.5 embryos were isolated seven days after PMSG/HCG injections, and hatched blastocysts without zona pellucida were used. CUT&RUN was performed as described above, with significant modifications, using adaptations described by Saiz et al.[49] for immunostainings. Embryos were manipulated in 1% agar-coated 4-well dishes (ThermoFisher scientific, Nunc 4-Well Dishes for IVF) throughout the protocol. Briefly, blastocysts were fixed using 400 μl of 0.1% formaldehyde prepared in PBS for 10 mins at room temperature, protected from light. After 10 mins, 40 μl of 1.42 M glycine was added to quench PFA and embryos were incubated at room temperature for 5 mins after gentle mixing, following which they were washed once in PBS and once in digitonin buffer, by sequentially passing them through wells containing the different buffers. Embryos were then incubated in 300 μl of antibody buffer (GATA6 or NANOG at 1:50) on a nutator at 4 °C for 1.5 h, following which they were washed thrice in digitonin buffer and then incubated in 300 μl of pA/G-MNAse solution. After incubation in pA/G-MNAse, embryos were again washed thrice and then collected in a 1.5 ml microfuge tube containing 150 μl of digitonin buffer. After incubating at 0 °C for 10 mins, MNase was activated as described in the CUT&RUN section. The rest of the protocol was followed as described for cells. Libraries were prepared using 20 amplification cycles. For the GATA6 at E3.5, CUT&RUN was performed on 120 pooled E3.5 blastocysts (first replicate), and separately on 160 pooled E3.5 s (second replicate) (depicted in Fig. 4e, f). At E4.5, GATA6 CUT&RUN was performed on a single replicate containing 55 pooled blastocysts. NANOG CUT&RUN was also performed on a single replicate containing 130 pooled E3.5 blastocysts.

**CUT&TAG**. CUT&TAG was performed, in duplicates, as previously described[39,81] with small modifications. mES cells were dissociated using Accutase (Sigma), counted, and 100,000 cells/replicate were pelleted at 600 g for 3 min at room

temperature. Cells were resuspended in Wash Buffer (20 mM HEPES pH 7.5, 150 mM NaCl, 0.5 mM Spermidine, 1x Protease inhibitor cocktail), and pelleted at 600 g for 3 min at room temperature. BioMag® Plus Concanavalin A beads (Bangs Laboratories) were equilibrated in Binding Buffer (20 mM HEPES pH 7.5, 10 mM KCl, 1 mM CaCl₂, 1 mM MnCl₂). Cells were resuspended in Wash Buffer, mixed with a slurry of equilibrated Concavalin A coated magnetic beads, and rotated for 10 min at room temperature. Per 100,000 cells, 10 µl bead slurry was used. Beads were resuspended in Wash Buffer containing 2 mM EDTA, 0.1% bovine serum albumin, 0.05% Digitonin, and 1:50 dilution of primary antibody and incubated on a nutating platform for 2 h at room temperature. After incubation, beads were washed twice in Digitonin Buffer (20 mM HEPES pH 7.5, 150 mM NaCl, 0.5 mM Spermidine, 1x Protease inhibitor cocktail, 0.05% Digitonin), then incubated with pA-Tn5 (6.7 uM, 1:100) in Dig-300 Buffer (20 mM HEPES pH 7.5, 450 mM NaCl, 0.5 mM Spermidine, 1x Protease inhibitor cocktail, 0.01% Digitonin) for 1 h at room temperature. After incubation, beads were washed twice, resuspended in 300 µl of Dig-300 Buffer containing 1 mM MgCl₂, and incubated for 1 h at 37 °C. After incubation, 10 µl 0.5M EDTA (Sigma), 3 µl 10% SDS, and 2.5 µl 20 mg/ml proteinase K (NEB) was added. Solutions were vortexed and incubated for 1 h at 50 °C, shaking at 900 rpm. After incubation, DNA separation and purification were performed as in CUT&RUN. DNA was finally eluted in 25 µl. To amplify libraries, DNA (20 µl) was mixed with 5X Q5 reaction buffer (10 µl), Q5 polymerase (0.5 µl), dNTPs (1 µl), nuclease free water (16 µl), and an equimolar mixture of a universal i5 and a uniquely barcoded i7 primer (2.5 µl), using different barcodes for each sample. The sample was placed in a Thermocycler with a heated lid using the following cycling conditions: 72 °C for 5 min (gap filling); 98 °C for 30 s; 14 cycles of 98 °C for 10 s and 63 °C for 30 s; final extension at 72 °C for 1 min and hold at 8 °C. Post-PCR clean-up was performed with a SPRIselect bead 0.6X left/1x right double size selection then washed twice gently in 80% ethanol and eluted in 10–12 µl 10 mM Tris pH 8.0. Multiplexed libraries were pooled and paired-end sequenced (2 × 50 bp) on an Illumina HiSeq2500. 1:50 dilutions of primary antibodies against H3K4me3 (Active Motif, 39159), H3K27ac (Abcam, ab4729), H3K27me3 (Cell Signaling, 9733 T) were used.

**Hi-C and Capture-C**. Hi-C (4dnucleome.org protocols) and Capture-C[82] libraries were prepared from two replicates following previously published protocols with minor modifications. Briefly, 1 million cells per sample were trypsinized, washed in growth media and fixed for 10 min at room temperature while rotating with 1% formaldehyde (Thermo: 28908) in 1 ml of HBSS media. To stop fixation, Glycine was added at final concentration of 0.13 M and incubated for 5 min at RT and 15 min on ice. Cells were then washed once in cold PBS, centrifuged at 2500 g 4 ºC for 5 mins (these centrifugation conditions were used for all washes following fixation) and pellets frozen at −80 ºC. Thawed cell pellets were incubated in 1 ml lysis buffer (10 mM Tris-HCL pH8, 10 mM NaCl, 0.2% Igepal CA-630, Roche Complete EDTA-free Sigma #11836170001). Following lysis, cells were dounced for a total of 40 strokes with a "tight pestle" and then washed in cold PBS. For DpnII digest, cells were resuspended in 50 µl 0.5% SDS and incubated at 62 °C for 10 min. Then 150 µl of 1.5% Triton-X was added and cells incubated for 15 min at 37 ºC while shaking at 900 rpm. 25 µl of 10X DpnII restriction buffer (NEB) was added, and cells further incubated for 15 min while shaking. 200 U of DpnII (NEB R0543M) were then added and incubated for 2 h, then 200 U more were added and incubated overnight. Next morning 200 U more were added and incubated for 3 h (total 600 U of DpnII). DpnII was inactivated at 62 ºC for 20 min. Biotin fill-in was done by incubating cells with a mixture of 4.5 µl dCTP, dTTP, and dGTP at 3.3 mM, 8 µl klenow polymerase (NEB M0210L) and 37.5 µl Biotin-14-dATP (Thermo 19524016) for 4 h at RT while shaking at 900 rpm for 10 s every 5 min. Ligation was done overnight at 16 °C also rotating at 900 rpm for 10 s every 5 min by adding 120 µl of 10X ligation buffer (NEB), 664 µl water, 100 µl 10% Triton-X, 6 µl BSA 20 mg/ml, and 2 µl T4 ligase (NEB cat #M0202M). For Capture-C, biotin fill-in step was skipped and 50 µl more of water was added to the ligation mix. Crosslink removal was done overnight with 50 µl of proteinase K in 300 µl of following buffer (10 mM Tris-HCl pH8.0, 0.5 M NaCl, 1%SDS) while shaking at 1400 rpm at 65 °C. Following Sodium Acetate and 100% Ethanol −80ºC precipitation, DNA was resuspended in 50 µl 10 mM Tris-HCl for Hi-C or 130 µl for Capture-C. Sonication for Hi-C was done using Covaris onetube-10 AFA strips using the following parameters for a 300 bp fragment size (Duration: 10 s, repeat for 12 times, total time 120 s, peak power-20 W, duty factor 40%, CPB-50). Sonication for Capture-C was done using Covaris AFA microtubes 130 with following settings for a fragment size of 200 bp fragments (Duration: 225 s, peak power-75W, duty factor 25%, Cycles per Burst-1000). Sonication was performed in a Covaris ME220 sonicator. Sonicated material was then size selected using SPRI beads with the following ratios: 0.55X and 1X for Capture-C and 0.55X and 0.75X for Hi-C. Hi-C material was then bound to 150µl Streptavidin C1 beads (Thermo 65002), washed and recovered, following manufacturers recommendations. Bead-bound DNA was resuspended in 50 µl 10 mM Tris HCl. Library preparation was done using Kapa Hyper Prep KK8502 kit. 10 µl of End-repair buffer and enzyme mix were added to resuspended beads and incubated for 30 min at RT and then 30 min at 65 °C. 1 µl of 15 mM annealed-Illumina adaptors, containing a universal p5 and an indexed p7 oligo, were then incubated with a mixture containing 40 µl of ligase and ligation buffer at RT for 60 min. Libraries were then amplified using 4 reactions per sample for a total of 200 µl and 10 cycles, as recommended by

manufacturer. For Capture-C, following sonication and size selection, 1 µg of template material was resuspended in 50 µl of 10 mM Tris and used for library prep with 10 µl of End-Repair reaction. 5 µl of 15 mM annealed -Illumina adaptors were ligated to the Capture-C material. Using a total volume of 100 µl, library was amplified by PCR using 6 cycles. For capture, 1 µg of Capture-C library per sample was mixed with mouse COT1 DNA and universal as well as index-specific blocking oligos from SeqCap EZ HE-oligo (Roche). 4.5 µl pool of biotinylated probes (xGen Lockdown Probe Pools from IDT), with each probe at 0.4 fmol/µl targeting the promoters of our loci of interest (sequences of each probe are listed in Supplementary Data 1h) were added to this mixture and incubated for 3 days at 47 °C. Following binding to Streptavidin C1 beads, material was washed as recommended by the SeqCap EZ Hybridization and Wash Kits. Following washes material was amplified by PCR using Kapa polymerase and 14 cycles. Material from different samples was then combined and 1 µg of pooled libraries was recaptured in a single reaction and amplified with 8 cycles.

**ChIP-seq**. ChIP-seq was performed, in duplicates, using the ENCODE protocol [www.encodeproject.org ChIP-seq_Protocol_v011014] with few modifications. Cells were harvested by trypsinization and counted. Approximately 30 million cells were fixed by resuspending them in 1% formaldehyde in HBSS (1 ml/million cells). Cells were incubated for 10 mins at room temperature, protected from light, on a nutator. After 10 mins, 1.42 M glycine (100 µl/ml of fixative) was added and cells were mixed by inversion and incubated for 5 mins at room temperature to quench PFA. Cells were then pelleted at 2000 g, at 4 °C for 5 mins and washed twice in 1X TBS (50 mM Tris-Cl, pH 7.5, 150 mM NaCl). Cell pellets were then flash frozen and stored at −80 °C. To perform ChIP, pellets were thawed on ice for 10 mins and lysed in Farnham lysis buffer (5 mM PIPES PH 8.0, 85 mM KCl, 0.5% NP-40, supplemented with protease inhibitors). To lyse cells, pellets were reconstituted in 1 ml buffer/10 million cells and incubated on ice for 10 mins, following which they were downced and then centrifuged at 2000 g, at 4 °C for 5 mins. The pelleted nuclei were then resuspended in 500 µl RIPA (50 mM Tris-HCl pH 8.0, 150 mM NaCl, 2 mM EDTA pH8, 1% NP-40, 0.5% Sodium Deoxycholate, 0.1% SDS, supplemented with ROCHE Complete protease inhibitor tablets (without EDTA), mixed gently by pipetting, and incubated on ice for 10 mins to achieve complete lysis. Extracted chromatin was then sonicated using Bioruptor (10 cycles of 30' on and 30', paused on ice for 5 mins, repeated 10 cycles of 30' on and 30'). Sonicated material was then spun down at 20,000 g, at 4 °C for 15 mins. Supernatant was collected in fresh tubes and chromatin was quantified using Qubit high sensitivity DNA kit. 10 µg of chromatin was used for each ChIP reaction, in a total reaction volume of 1 ml, adjusted with RIPA. For each sample, 30 µl of Protein A/G beads preconjugated to 2 µg antibody (by incubation of antibody with washed beads resuspended in 500 µl RIPA for 3 h) was added to the reaction and chromatin was capture by overnight incubation at 4 °C, on a rotating platform. The next day, captured chromatin was washed successively as follows: once in low-salt wash buffer (0.1% SDS, 1% Triton X-100, 2 mM EDTA, 20 mM Tris-HCl pH 8.0, 150 mM NaCl), twice in high-salt wash buffer (0.1% SDS, 1% Triton X-100, 2 mM EDTA, 20 mM Tris-HCl pH 8.0, 500 mM NaCl), twice in lithium chloride wash buffer (250 mM LiCl, 1% NP-40, 1% Sodium Deoxycholate, 1 mM EDTA, 10 mM Tris-HCl pH 8.0), and twice in 1X TE (10 mM Tris-HCl pH 8.0, 1 mM EDTA). Bound chromatin was then eluted in 200 µl freshly prepared direct elution buffer (10 mM Tris-HCl pH8, 0.3 M NaCl, 5 mM EDTA pH8, 0.5% SDS) supplemented with 2 µl RNAse A (10 mg/ml), by incubating on a thermomixer at 65 °C, at 800 rpm, overnight. The next day, beads were discarded, and the supernatant was incubated with 3 µl of Proteinase K (20 mg/ml) at 55 °C, 1200 rpm, for 2 h to reverse crosslinks. DNA was then eluted in 20 µl deionized water using the Zymo DNA Clean and Concentrator kit (Zymo, D4033). 10 µl of eluted DNA was used for library preparation exactly as described for CUT&RUN. 12 amplification cycles were used per sample, and library was sequenced on HiSeq2500 using PE50.

**Sequential ChIP-seq (reChip)**. Cell-fixation was carried out as described above for ChIP-seq. After fixation cells were frozen as pellets of 10 to 20 million cells per tubes. For the first round of ChIP, fixed pellets of undifferentiated cells and cells induced with Dox for 2 h were thawed on ice for 10 mins. For each time point 6 replicates of 10 million cells each, were used. Nuclei were extracted and lysed from thawed cell pellets using the Covaris truCHIP® Chromatin Shearing Kit, following recommendations for high cell numbers (1 ml lysis buffer/20 million cells). Lysed nuclei were sonicated in Covaris 1 ml AFA Fiber milliTUBE (Duration: 2100Secs, peak power-75W, duty factor 15%) to fragment chromatin on an average size of 200 bp. Sonicated chromatin was centrifuged at 20,800 g for 15 mins to separate residual debris and the cleared lysate was transferred to a fresh pre-chilled tube. 25 µl was aliquoted separately to confirm sonication efficiency and estimate chromatin concentration while the rest was frozen and stored at −80 °C. To the 25 µl, 1 µl of RNase A (10 mg/mL) was added and incubated at 37 °C for 30 min after which 1 µl of Proteinase K (10 mg/mL) was added and chromatin was reverse cross-linked overnight at 65 °C in a PCR cycler with a heated lid. The following day, after performing 2X SPRI bead clean fragment size was confirmed by running 1 ng on a tape station, and concentration was estimated using Qubit 1X dsDNA high sensitivity kit. Frozen chromatin was thawed and NaCl (final concentration of 150 mM), Triton-x-100 (final concentration of 1%) and Na-deoxycholate (final concentration of 0.1%) were added. 10 µg chromatin was used per replicate and a

total of six replicates for each time point were subjected to immunoprecipitation with anti-FLAG M2 (Sigma, F3165-2MG) conjugated to protein A/G beads as described for ChIP. Volume was made up to 1 ml using freshly prepared RIPA. After overnight ChIP washes were performed as described in ChIP. To elute FLAG-bound chromatin, beads were resuspended in 60 μl of freshly made Direct Elution Buffer (10 mM Tris-HCl pH8, 0.3 M NaCl, 5 mM EDTA pH8, 0.5% SDS) supplemented with 10 mM DTT and incubated at 37 °C, 500 rpm, 15 mins. Eluted material was collected in a fresh tube and the process was repeated once more and pooled with the first elution. Finally, eluates from three replicates were pooled together to give two replicates per timepoint. 100 μl was subjected to reverse crosslinking and library preparation to map FLAG-bound regions, and 100μl was added to 900 μl RIPA and used for immunoprecipitation with anti-Nanog conjugated to protein A/G beads and the protocol described under ChIP was followed. 20 amplification cycles were used to prepare libraries.

**RNA-seq data analysis**. RNA-seq analysis including principal component analysis and identification of differentially expressed genes was performed using LCDB workflow [github.com/lcdb/lcdb-wf]. Briefly, the data was first assessed for quality control using fastQC v0.11.8 (bioinformatics.babraham.ac.uk/projects/fastqc/) to look for sequencing quality issues and trimmed for quality using cutadapt v2.7[83]. No significant quality issues were detected. The RNA-Seq data was then mapped to the mouse genome version GRCm38.p6 using HISAT2 v2.1.0[84]. Stats for number of reads and peaks can be found in Supplementary Data 1. Expression levels were estimated at gene-level with featureCounts (subread v1.6.4)[85] using the GENCODE version m18 annotation. Differential expression analysis was performed using DESeq2 v1.22.1[86] using the 'normal' log-fold-change shrinkage. Statistical significance was defined as false discovery rate (FDR) < 0.1 and |log2FoldChange| >= 2 (4-fold difference). The count data was also normalized using the variance stabilizing transformation for use in visualization purposes. Genes with a log2fold change higher than 2 or lower than −2, and adjusted p value lower than 0.1 were identified as differentially expressed. Each time point was compared to previous time point and to 0 h to identify all genes differentially expressed during differentiation. For plotting transcript levels at different datapoints for selected genes, DE-Seq2 normalized values were used. DE-Seq2 normalized values were also used to plot changes in gene expression during differentiation in heatmaps using row zscores for each gene calculated in R. We also calculated the level of expression of the transgene compared to the endogenous *Gata6* locus. The *Gata6* transgene encodes only the coding sequence of *Gata6*. Thus, reads coming from the coding region of the gene originate both from endogenous *Gata6* and the transgene, while untranslated regions (e.g. 3'UTR) are expressed only by endogenous *Gata6*. We utilized this information to calculate the expression levels of the transgene. First, we calculated the number of reads mapping to the 3'UTR and the CDS in the last exon of *Gata6*. The counts were normalized to library size (per million) and sequence length (per kb) to avoid sequencing depth and feature length biases. The transgene expression, $T$, was then calculated as the difference between the normalized CDS and UTR expression, while endogenous expression, $E$, was taken to be the mean of the normalized 3'UTR expression at 96 h ($n = 4$). Finally, the relative expression of the transgene was calculated as $T/E$.

**Single-cell RNA-seq analysis**. Publicly available single-cell RNA-Seq data from[22] was downloaded as a cell x count matrix. The cell clustering determined in the study was downloaded and the count matrix was subset to only the relevant timepoints (embryonic day E3.5 and E4.5). For use in our study we renamed and combined some clusters based on cell identities as EPI, PrE or ICM: E3.5:1 = E3.5_ICM, E3.5:0 + E3.5:4 = E3.5_PrE, E3.5:2 = E3.5_EPI-E4.5:0 + E4.5:1 = E4.5_PrE, E4.5:2 = E4.5_EPI, E4.5:3 = E4.5_TE. Next, we reanalyzed the data using the Seurat R package v3.1.2 using the standard workflow to find conserved markers and differentially expressed markers. For differential expressed markers we specifically compared PrE and EPI cells at each time-point (E3.5 and E4.5) and also E4.5_PrE and E3.5_EPI. The list of differentially expressed genes between PrE and Epi was also compared to the list reported by Mohammed and colleagues[21] and only genes present in both datasets was kept for further analysis. Next, we compared the differentially expressed markers between E3.5 and E4.5 to the genes that are different between the sorted and bulk populations. To perform a comparison between bulk and single-cell RNA-Seq data, we performed the following procedure. First, we extracted single-cell counts from cells belonging to E3.5 and E4.5 stages based on cell metadata. Next, we normalize the data to library size (per million) at cell level and performed log transformation after adding pseudocount of 1 (log1p). This is similar to the logNormalize procedure implemented in the Seurat R package. We then averaged cells within each representative cluster. This yielded mean normalized data corresponding to each cell stage: E3.5 – ICM, PrE, EPI; E4.5 – PrE, EPI, TE. For bulk RNA-Seq, we performed variance stabilizing transformation from the DESeq2 R package (v1.22.1) to normalize count data and calculated a mean across biological replicates for each time-point. Next, for comparing such disparate sets of data, the technical variability between bulk and single-cell data needed to be regressed out. To this end, we then fit a model: *expression ~ group*, where the factor *group* was either *single-cell* or *bulk*. The residuals from this modeling procedure was then used to generate a PCA plot.

**ATAC-seq analysis**. The ENCODE ATAC-seq pipeline [github.com/ENCODE-DCC/atac-seq-pipeline] was used to process ATAC-seq and to obtain bigwig files for visualization in IGV and to produce heatmaps or signal profiles using Deep-Tools (v3.5.1). For this, the bigwig files generated using signal p value and with the 2 replicates pooled were chosen. For downstream analysis, we used the IDR-identified peaks for each condition calculated from the 2 available replicates at each time point as these were the most stringent peaks identified. Diffbind (v2.12.0)[87] was used to plot PCA and identify peaks with differential accessibility during differentiation by comparing 48 h to 0 h. Stats for number of reads and peaks can be found in Supplementary Data 1. To compare ATAC-seq between bulk GATA6-induced cells and a PDGFRA$^+$-sorted population we also used Diffbind. No peak was identified as differential between sorted and bulk populations. For both types of comparisons, the DESEQ2 option was used and an adjusted pvalue of 0.0001 and log2 fold change of 2(or −2) was used as threshold of significance. All timepoints were compared against 0 h. ATAC peaks were classified into stable, increased, or decreased by comparing the 0 h and 48 h time-points. Differential ATAC peaks were identified with DiffBind. We used GREAT[88] (v4.0.4) to further identify which of these peaks were associated with Epi or PrE specific genes. For this, peaks were assigned to a single gene if it was within a maximum distance of up to 50 kb. A combination of Deeptools[89] and bedtools (v2.30.0)[90] was used for plotting heatmaps and summary profiles. Deeptools profiles and heatmaps were plotted using peak center as the single reference point, profiles were plotted using mean signal. Epi and PrE CREs depicted in Fig. 1d are reported in Source Data. CREs identified by our approach were compared to the ENCODE CRE database and overlapping CREs are also reported in Source Data. ATAC-seq nucleosomal signal over TF motifs was plotted using the ATACseqQC (v3.15)[91] pipeline by merging both replicates for the same time point. Only nucleosomal signal (fragments larger than 180 bp) is shown. We used FIMO[92] (vMEME 5.4.1) to identify TF motifs at peaks of interest identified by CUT&RUN. FIMO was run using default settings and a q-value threshold of 0.001 and with the following JASPAR motifs: GATA6 (PB0023.1), NANOG (UN0383.1). To plot ATAC footprinting signal over HNF1B motifs (MA0153.1) we used the TOBIAS pipeline [https://github.com/loosolab/TOBIAS][93] with merged ATAC-seq replicates. ATAC signal was plotted over all motifs identified as bound at 48 h by this pipeline.

**CUT&RUN and CUT&TAG analysis**. Paired-end 50 bp reads were processed using Bowtie2 (v2.4.5)[94], with the following options (-N 1 --local --very-sensitive-local --no-unal --no-mixed --no-discordant --phred33 -I 10 -X 700 -x). Reads that mapped to ENCODE mm10 blacklist regions were removed using Samtools (v1.15)[95]. Piccard (v2.27.2) (broadinstitute.github.io/picard/) was then used to identify non-duplicated reads. Duplicate reads were removed for libraries generated from blastocysts but kept for all others. Diffbind was used to analyze replicates by plotting pearson correlation of signal found over identified peaks (Supplementary Fig. 8). For IGV visualization, and plotting heatmaps and profiles, Deeptools v3.5.1 was used and signal of replicates was merged and normalized. In the case of CUT&RUN of Epi TFs and CUT&TAG, signal normalization was done using using total read count. For Epi TFs, signal normalization was done with reads that mapped to *E.coli* and *S.cereviseae* genomes. Stats for number of reads and peaks can be found in Supplementary Data 1. For TF signal, only fragments smaller than 120 bp were used while for histone modifications, fragments larger than 150 bp were selected. MACS (v2.1.1.20160309) was used to identify regions of enrichment using the narrow option for TFs and the broad option for histone modifications. Peak calling was done using as a control CUT&RUN samples generated using rabbit anti-rabbit antibody as a control. Peaks overlapping with ENCODE blacklist regions were removed from further downstream analysis. A combination of deeptools (v3.5.1) and bedtools (v2.30.0) was used for plotting heatmaps and summary profiles. Characterization of GATA6 peak types was done considering the top 10,000 peaks at each time point according to adjusted *q* value calculated by MACS. This valued was chosen based on previous comparison to published GATA6 ChIP-seq data[35]. Peaks were considered as Early if present in the top 10,000 peaks at any timpoint between 2 and 8 h for a total of 14,906 early peaks. If present only at 48 h peaks were identified as Late (total 5850). Early peaks that intersected with ATAC peaks at 0 h were considered as Early Open at 0 h and the remaining Early peaks were considered Early Closed at 0 h. These peaks were then associated with PrE or Epi genes using the same approach as for ATAC-seq. Peaks were further divided into proximal or distal depending on distance to TSS (less than 5 kb away were considered proximal). NANOG peaks were identified similarly to GATA6 and the top 15,000 peaks selected for characterization based on number of peaks detected by ENCODE data. To define which peaks overlapped with GATA6, all GATA6 peaks identified from 2, 4 and 8 h were used. NANOG peaks were classified as Epi or PrE specific as done for ATACseq. For analysis of NANOG peaks that appear specifically at 2 h, all peaks were considered and separated into 2 clusters. Peaks that overlapped with NANOG peaks already identified at 0 h and peaks that were only identified at 2 h. These were separated into those that overlapped with ATAC-seq peaks at 0 h and those that didn't. CUT&RUN analysis of Cutsite probability was performed using CUTRUNTools[96] using the GATA6 (PB0023.1) motifs identified by FIMO (vMEME 5.4.1) over the mentioned peaks in figure. A threshold of 0.001 was used to identify motifs within peaks.

**TF motif analysis**. To identify TF motifs for GATA6 and OCT4/SOX2 at peaks of interest we used FIMO (vMEME 5.4.1) with a pvalue threshold of 0.001 and with the following JASPAR motifs: GATA6 (PB0023.1), and OCT4/SOX2 (MA0142.1). Then, for each peak we calculated the number of motifs found in each peak and displayed this number as a boxplot. To assess motif strength, for all motifs identified with a pvalue lower than 0.001, we plotted the FIMO-calculated qvalue. For comparison of motifs in two different conditions we used Homer to identify known motifs enriched in one condition versus another as shown in figure.

**Hi-C and Capture-C analysis**. Hi-C libraries were sequenced with paired-end reads of 51 nucleotides. Data was processed using the Hi-Cpro pipeline (v3.1.0)[97] to produce a list of valid interactions pairs. This list was converted into cool and mcool files for visualization with higlass (v1.11.7)[98]. Stats for number of reads and peaks can be found in Supplementary Data 1. Eigenvalues were calculated using homer (v4.11.1)[99] with 250 kb bins and the runhicpca.pl script. For comparison between conditions, we used the gethiccorrDiff.pl script also at 250 kb resolution. Bins were considered as belonging to different compartments if correlation coefficient was lower than 0.4. For heatmaps containing eigenvalues, bins were sorted from highest eigenvalue to lowest based on the 0 h score. H3K27me3 and H3K27ac score for each bin was determined using Deeptools (v3.5.1). Capture-C were libraries were also sequenced and processed the same way. The make_viewpoints Hicpro script was used to obtain individual Capture-C bigwig files for each replicate of each viewpoint with 1 kb-sized bins. For visualization, averages of each replicate were used. $P$ values were calculated using DESeq2 (v1.22.1) and comparing Capture-C signal of 2 replicates over overlapping 5 kb windows across the regions shown in figure. Overlapping 5 kb windows were built by sliding each window 500 bps. Comparisons were done between 0 h and either 16 h or 48 h. Adjusted p values lower than 0.01 were considered as threshold for significance. Horizontal bars represent windows considered as statistically significantly different for these comparisons.

**Statistics & reproducibility**. The number of replicates used in each assay and statistics used in analysis of the different datasets are described under the corresponding subsections in methods. No statistical method was used to predetermine sample size. No data were excluded from the analyses except for samples where QC failed.

**Reporting summary**. Further information on research design is available in the Nature Research Reporting Summary linked to this article.

## Data availability

The datasets generated in this study can be accessed on the Gene Expression Omnibus database under the accession number GSE181104. Source data are provided with this paper.

## Code availability

All software used for analysis is freely available and can be found in the methods section. For RNAseq analysis code can be found under the LCDB workflow [github.com/lcdb/lcdb-wf]. The ENCODE ATAC-seq pipeline was accessed from [github.com/ENCODE-DCC/atac-seq-pipeline]. GREAT analysis was performed on [http://great.stanford.edu/public/html/]. To determine TF occupancy based on ATAC-signal, TOBIAS pipeline [https://github.com/loosolab/TOBIAS] was run. Prediction of Cutsite probability from CUT&RUN data was determined using CUTRUNTools [https://github.com/CutRunTag-jusuE404/cutruntools/blob/master/README.md].

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

## Acknowledgements
We thank all members of the Unit on Genome Structure and Regulation for discussions and Nestor Saiz, Effie Apostolou, Karl Pfeifer, Sergio Ruiz, and Judith Kassis for comments on the manuscript. We thank NICHD's molecular genetics core, specifically Steven Coon, Tianwei Li and James Iben. This work utilized the computational resources of the NIH HPC Biowulf cluster (http://hpc.nih.gov). We thank Lisa Price for cell sorting experiments. We thank the mouse core of NICHD specifically Jeanne Yimdjo, Victoria Gibbs and Alexander Grinberg. We are also indebted to Kathy Niakan for providing the inducible GATA6 mES cells and Steven Henikoff for pA-MNase and pA-Tn5 enzymes. This work was funded by NIH intramural project HD008975-02 (PPR) and HD008986-03 (RKD).

## Author contributions
J.T., D.L., S.F. and P.P.R. performed experiments. J.T., A.M., and P.P.R. analyzed experiments. R.K.D. and P.P.R. supervised the project. J.T. and P.P.R. designed the project and wrote the paper with input from all other authors.

## Funding

## Competing interests
The authors declare no competing interests.
