## [Peer Review File · Nature Communications]

Extensive co-binding and rapid redistribution of NANOG and GATA6 during emergence of divergent lineagesREVIEWER COMMENTS

Reviewer #1 (Remarks to the Author):

I read with great interest the manuscript by Thompson et al. entitled “Rapid redistribution and extensive binding of NANOG and GATA6 at shared regulatory elements underlie specification of divergent cell fates”. In this manuscript the authors use a Gata6 inducible overexpression system to globally characterize the molecular mechanisms whereby this transcription factor controls the differentiation of ESC into primitive endoderm (PrE). By generating extensive and high-quality genomic data, the authors uncover that GATA6 acts as a pioneer TF during the activation of regulatory elements associated with PrE genes. In addition, they also find that GATA6 binds to regulatory elements linked to epiblast (Epi) genes together with major pluripotency TFs (e.g. NANOG, SOX2, OCT4), potentially facilitating their silencing/decommissioning. Interestingly, the binding of GATA6 to these Epi-specific regulatory elements seems to evict pluripotency TFs (NANOG and SOX2), which can then transiently bind to PrE-specific regulatory elements bound by GATA6. Based on these observations, the authors propose that the extensive binding of PrE and Epi-specific TFs to bind and potentially regulate a shared transcriptional network might confer the inner cell mass with its differentiation plasticity. These findings can provide important insights into the molecular mechanisms whereby TFs with master regulatory functions control lineage specification and the differentiation of progenitor cells. Consequently, it can be of broad interest to scientists interested in developmental biology and/or gene expression control.

Overall, the presented data seems to be of high quality and it supports the authors' claims. My only major concern is that the authors do not perform experiments to better understand the relevance and mechanistic basis of their most interesting observations. Consequently, the most novel aspects of the manuscript are, in my opinion, largely based on correlative genomic data. Importantly, the authors are quite careful in their conclusions and do not make overclaims. These are some suggestions that I think could help the authors to improve the manuscript:

1. I think the authors do not sufficiently explore why GATA6 seems to act as pioneer TF for some but not all PrE-specific CREs. For example, did the authors find any overrepresented motifs among the early CREs that GATA6 can pioneer?. Do these early sites show any particular chromatin features?. Does GATA6 pioneer these early sites alone or cooperatively with other TFs (e.g. pluripotency TFs such as OCT4 or SOX2)?.
2. The fact that GATA6 transiently binds Epi-specific CREs and potentially helps to silence them is very interesting. However, the authors barely describe how GATA6 might preferentially act as an activator in the context of PrE-CREs and as repressor for Epi-CREs. What is the mechanism whereby GATA6 can act as a repressor?. Do GATA6 binding motifs overlap with NANOG motifs within Epi-CREs?. If this is the case, GATA6 and NANOG could compete for binding to nearby sites, with GATA6 acting as a short-range repressor. Alternatively, GATA6 could recruit co-repressors (e.g. HDACs).
3. In order to provide more direct evidences supporting that GATA6 directly represses Epi-specific TFs by binding to and silencing their associated CREs, the authors could use

CRISPRs to delete/mutate GATA6 binding motifs within some of those CREs. If GATA6 binding to those CREs is important for their silencing, then the deletion of GATA6 sites should reduce silencing.

4. Another interesting observation is that pluripotency TFs transiently bind to PrE CREs bound by GATA6. However, it is rather unclear whether the binding of the pluripotency TFs to those CREs is of any functional relevance. Are these TFs repressing, activating or simply binding to those CREs?. As suggested in the previous point, the authors could delete/mutate binding sites for some of these pluripotency TFs in order to address their functional relevance.

5. The authors show binding of GATA6 and Epi-specific TFs to a large set of shared CREs. Do the authors think that all these TFs simultaneously bind to the same CREs within the same cells and alleles?. It is possible that, upon GATA6 overexpression, there is increased cellular heterogeneity, with cells expressing variable levels of GATA6 and other TFs. Therefore, the authors could perform sequential ChIPs (reChIP) to more conclusively show whether GATA6 and pluripotency TFs simultaneously bind to PrE and Epi-specific CREs during early induction time points.

Minor points:

1. The authors should investigate GATA6 protein levels by western blot and/or immunofluorescence at various time points after induction.

2. The authors claim that at many PrE-specific regulatory elements, increased accessibility precedes transcriptional changes of the target genes. However, gene expression was investigated by polyA RNA-seq. The authors should investigate for at least some genes nascent transcripts rather than mRNA in order to better support their claims.

3. The lack of H3K27me3 and H3K9me3 at PrE-specific CREs activated by GATA6 could be perhaps explained by the overall low levels of these histone marks in ESC cultured under 2i conditions. Moreover, these histone marks are not prevalent at distal CREs. The authors could use ChIP-seq (or similar) data generated in ICM for these histone marks to better support their findings.

Reviewer #2 (Remarks to the Author):

In this manuscript, Thompson et al. use GATA6-driven differentiation of mES cells to examine plasticity in cells from the inner cell mass. Specifically, the authors are interested in PrE vs Epi cell fates and how cis regulatory element regulation is altered through chromatin accessibility and factor localization to drive these cell fates. The authors perform multiple genomic assays (RNAseq, ATAC-seq, CUT&RUN, CUT&Tag, and Hi-C) to assess the environment at these cis regulatory elements and ultimately put forth a model on how NANOG and GATA6 act in this cell fate decision. Importantly, the authors also perform a few experiments in blastocysts

to support their findings. Overall, the insights gained from the manuscript are exciting, and would be of interest to the field, however I have the following concerns:

1) In some of the datasets, it is unclear what the authors are presenting: for example, in Figure 1D, are the accessible locations called from the 48 hour timepoint, the 0hr timepoint, or all the accessible regions in any time point? Pending how these were called, for example, if they were called based on 0hr vs 48 hr, then are there locations that are up/down in the other time points that are not represented?

a. Related, the authors show in S1 non Epi or Pre-specific locations with changes in accessibility. What are these locations and why would they be changing upon GATA6 transgene expression?

2) The authors often overstate their data, drawing conclusions that are models or hypotheses based on the data. Below, I am listing a few examples, but there are more throughout the manuscript.

a. For Figure 1, it is surprising the authors do not just show the accessibility over time of the genes that are up and downregulated and then correlate these findings with the expression data. This is especially important as the authors make a claim on page 6 line 99 “changes in chromatin accessibility were more striking than those observed at the transcriptome level”. This is an extremely subjective comment that is not grounded in presenting quantitative results. Does striking mean number of peaks? If so, this should be explicit and also demonstrated in a quantitative manner.

b. on page 6 line 111 “...possibly due to a change in binding of a Sox7 repressor. This is a good example of why not all PrE CREs gain accessibility as it may have been expected”. While the authors are pulling Sox7 as an interesting example, there is no mechanistic experiment demonstrating the repressor binding has changed, and therefore this is not an actual mechanistic example, but rather a qualitative finding that could match multiple mechanisms.

c. On Page 7 line 113-114 “These data demonstrate that GATA6-driven chromatin remodeling is initiated immediately following induction”. The authors have not shown that GATA6 drives chromatin remodeling, but rather transgene expression of GATA6 results in alter accessibility. Some of these changes may be indirect and a result of other changes occurring during differentiation.

3) Throughout the paper, the authors show a lot of their data qualitatively. It is never clear how many replicates were performed, the overlap in those replicates (in terms of correlation), correlation in the timepoints over GATA6 induction, and quantitative peak calling/intersecting peak information (which could only be performed with the appropriate negative controls). Some of this information is included in the S1 supplement, but overlapping peaks from different datasets and making it more clear within the manuscript would benefit readers.

4) Experimentally, the authors perform no negative controls for their extensive CUT&RUN/CUT&Tag experiments (either IgG or no primary antibody). This is concerning, as we do not know the background cutting in their experiments.

5) Relatedly, the profiles presented for some of the datasets are concerning. For example, in Figure 2B, the background cutting/transposition events in the same profiling groups, at different timepoints are very similar. While there may be some similarities in the background, these are alarmingly similar. Without a negative control, this is especially concerning. Further, without access to the data to analyze, reviewers cannot confirm these findings.

Minor comments:

- 1) In the presentation of Figure 1, the authors demonstrate genes that are up and downregulated over their timecourse of GATA6 induction. Then the authors examine CREs for PrE and Epi. As CREs typically mean promoters and other more distal regulatory elements, such as enhancers or silencer, how were these distal regulatory elements identified?
- 2) The K4me3 data presented in Figure 2B is very peculiar. It is perhaps due to these mostly being gene distal CREs, but if that is the case the authors should clarify. K4me3 over promoters should look much more consistent with prior findings of this pattern, and the authors might consider showing those data to demonstrate the experiment worked.
- 3) In Figure 2D it is surprising that there is no enrichment at all in the 0hr time point for GATA6 or K27ac, as free pA/G-MNase or Tn5 would cut/transpose to permit fragment release. This could be clarified by having the appropriate negative controls performed in parallel to the experiments.
- 4) It is lost on me how in Figure 2E, left panel, these are accessible peaks, when the 0hr time point has occupancy of nucleosomes, similar to the 0hr time point on the right (non-accessible) panel.
- 5) For Figure S2, the authors perform motif finding for peaks underlying the GATA6 locations. It is surprising that a) the authors show only one enriched motif and b) GATA6 motif was not identified. Were other significant motifs found? Further, later from the findings, we might have expected to see Oct-Sox-Nanog motifs; were they present?
- 6) On page 12 line 201-202 the authors state "Since Nanog levels decreased faster than Sox2 during PrE differentiation...". While this may be true, the slope for Nanog does look steeper than Sox2, this is not quantified and should be if the authors are going to make this statement. Also, given Nanog is more highly expressed, it should be normalized for expression.
- 7) When the authors profile NANOG and SOX2 occupancy over GATA6 induction, they show that SOX2 binding persists while NANOG occupancy is decreased/lost. Did the authors discuss why SOX2 might be persistent in binding?
- 8) For Figure 4A, the second and third rows have the same number of locations listed (9830). It is surprising to me that this is an exact match, and therefore I think this is a typo.
- 9) The authors performed blastocyst CUT&RUN and it is not clear if they used individual blastocysts or pooled blastocysts.

Reviewer #3 (Remarks to the Author):

The manuscript by Thompson and collaborators reports ES cell transdifferentiation upon ectopic induction of Gata6 expression. They focus on the dynamic changes of transcription factors bindings on potential control regions.

Such kind of experiments has previously been already carried out by Wamaitha et al (2015), but the present analysis is more thorough and, importantly examines early time points.

They found that chromatin changes occur as early as 2 hours after transgene induction. The binding to PrE CREs shows different dynamics (early, late) revealing different mechanisms. GATA6 also binds to Epi CREs, as previously shown (Wamaitha, 2015). Here, a systematic analysis of Epi specific transcription factors (TFs) binding such as NANOG, SOX2 was carried out in parallel. A high number of Epi CREs bound by Epi TFs are also bound by GATA6. This enables to show the dynamics of eviction of these factors from Epi CREs by GATA6. Interestingly they observe a transient de novo binding of Epi TFs on PrE CREs occupied by GATA6 most likely due to novel open chromatin sites. They observe a difference in density and affinity for binding sites for GATA6 or for SOX2/OCT4 between Epi and PrE CREs that could explain the differences in binding dynamics.

Altogether this is a thorough CRE binding analysis during GATA6 conversion of Epi cells into PrE, prolonging the findings by Wamaitha and collaborators.

There are several concerns about the manuscript:

1- there is no mention of cell heterogeneity, which is present in both embryo and cells. Several groups have shown that NANOG and GATA6 proteins have cell-to-cell heterogeneous expression in E3.25 embryos (Dietrich and Hiragi, Development 2007; Bessonard et al, Development 2014; Saiz et al, Nat Com 2016), and this is obvious in fig S4C. NANOG heterogeneity in ES cells has been reported by many labs, even cultured in 2i (Abranches et al, Development 2014, 10.1242/dev.108910). Cell heterogeneity was observed until at least 8h after induction using the same dox system (Wamaitha, 2015; Schroter, 2015).

The bulk RNAseq comparison with scRNAseq from embryonic cells is informative but a single cell analysis would have provided much more information on the cell states analysed.

While transcription factors binding in single cells is challenging, scATAC-seq is being used increasingly.

Authors have rightly used cell sorting with PECAM1 and PDGFRa, however only at 48h after induction. Using the FACS at each time point would be unable to regroup and analyse cells with a similar identity, as was done for example by LoNigro et al, Stem Cell Rep 2017 (10.1016/j.stemcr.2016.12.010).

Trying to eliminate cell heterogeneity in samples (or use sc technologies) is important. For example, the binding of Epi TF and GATA6 at the same CREs could be due to cell heterogeneity. While this will be difficult to solve, this limitation should be at least commented in the main text.

2- ES cells are most similar to an E4.5 epiblast cells and ES cells do not give rise to PrE cells in vivo. So what is induced here is transdifferentiation. Due to different starting cell states and chromatin landscapes, the mechanism to produce PrE cells is certainly different than in the embryo. ES cells cannot be used to study early time points of PrE differentiation. The analysis that was carried out on embryos is thus important (although difficult to interpret due to cell

heterogeneity). Nevertheless, it should be reminded in the main text that this artifactually induced transdifferentiation is certainly different than in vivo.

3- It is interesting to see that Gata6 RNA levels reach their maximum already after 2 hours of induction. However, are these levels (transgene and endogenous) physiological? What are these levels compared to in vivo.

4- GATA4 and SOX17 relay GATA6 on PrE CREs. Is it the case as well on Epi CREs?

5- Are there some of the CREs described here that were previously validated functionally (i.e., indeed control the target gene). This would enrich the data.

6- Fig 2E left panel: why does it look like a nucleosome is present at 0h while it is supposed to be accessible chromatin?

7- Lists of ranked genes/CREs should be provided for the different experiments (Figures 1C/S1D; 2B; 3B; 3E; 4A, 4F, 5B bottom part). As well, I guess that all data will be deposited to a publicly available repository, as nothing is mentioned in the manuscript.

Reviewer #1

I read with great interest the manuscript by Thompson et al. entitled “Rapid redistribution and extensive binding of NANOG and GATA6 at shared regulatory elements underlie specification of divergent cell fates”. In this manuscript the authors use a Gata6 inducible overexpression system to globally characterize the molecular mechanisms whereby this transcription factor controls the differentiation of ESC into primitive endoderm (PrE). By generating extensive and high-quality genomic data, the authors uncover that GATA6 acts as a pioneer TF during the activation of regulatory elements associated with PrE genes. In addition, they also find that GATA6 binds to regulatory elements linked to epiblast (Epi) genes together with major pluripotency TFs (e.g. NANOG, SOX2, OCT4), potentially facilitating their silencing/decommissioning. Interestingly, the binding of GATA6 to these Epi-specific regulatory elements seems to evict pluripotency TFs (NANOG and SOX2), which can then transiently bind to PrE-specific regulatory elements bound by GATA6. Based on these observations, the authors propose that the extensive binding of PrE and Epi-specific TFs to bind and potentially regulate a shared transcriptional network might confer the inner cell mass with its differentiation plasticity. These findings can provide important insights into the molecular mechanisms whereby TFs with master regulatory functions control lineage specification and the differentiation of progenitor cells. Consequently, it can be of broad interest to scientists interested in developmental biology and/or gene expression control. Overall, the presented data seems to be of high quality and it supports the authors’ claims. My only major concern is that the authors do not perform experiments to better understand the relevance and mechanistic basis of their most interesting observations. Consequently, the most novel aspects of the manuscript are, in my opinion, largely based on correlative genomic data. Importantly, the authors are quite careful in their conclusions and do not make overclaims. These are some suggestions that I think could help the authors to improve the manuscript:

1. I think the authors do not sufficiently explore why GATA6 seems to act as pioneer TF for some but not all PrE-specific CREs. For example, did the authors find any overrepresented motifs among the early CREs that GATA6 can pioneer? Do these early sites show any particular chromatin features? Does GATA6 pioneer these early sites alone or cooperatively with other TFs (e.g. pluripotency TFs such as OCT4 or SOX2)?

To characterize the difference between GATA6 target sites bound early (where it likely acts as a pioneer factor) versus late, we carried out motif analysis to identify 1) differences in GATA6-motif density and 2) if late GATA6 sites required the binding of additional factors as compared to early GATA6 target sites. We found that late GATA6-target sites contained fewer recognition motifs as compared to early GATA6-target sites, which could explain why they are bound later in differentiation once GATA6 is expressed at higher levels. This analysis can now be found in Fig. S2D. Additionally, we found that compared to 48h-specific GATA6 targets, the early GATA6-bound sites were enriched for the NR5A2 binding-motif. NR5A2 is a TF essential for pluripotency, which controls the binding of NANOG, SOX2, and OCT4. (see for example PMID: 34397088). We propose that, as we describe for NANOG and GATA6, NR5A2 may also display GATA6-mediated reshuffling to promote quick PrE program activation. We have included this result in Fig S2E. We did not identify OCT4-SOX2 or other PrE TFs motifs as significantly enriched.

2. The fact that GATA6 transiently binds Epi-specific CREs and potentially helps to silence them is very interesting. However, the authors barely describe how GATA6 might preferentially act as an activator in the context of PrE-CREs and as repressor for Epi-CREs. What is the mechanism whereby GATA6 can act as a repressor? Do GATA6 binding motifs overlap with NANOG motifs within Epi-CREs?. If this is the case, GATA6 and NANOG could compete for binding to nearby sites, with GATA6 acting as a short-range repressor. Alternatively, GATA6 could recruit co-repressors (e.g. HDACs).

In line with the reviewers comments, in the last two paragraphs of our third results section we also wondered about how GATA6 may repress Epi CREs. Our data suggested that a difference in motif densities could impact whether GATA6 functions as an activator or repressor. Analysis of GATA6 motif density in PrE versus Epi sites, showed that PrE CREs contain higher density of high-affinity GATA6 recognition motifs compared to Epi CREs (Fig S3D). In our discussion, we described how differences in motif density could allow the same TF to function as an activator or repressor. We hypothesized that CREs carrying higher densities of strong TF motifs would be bound longer and more stably by the TF, marking them for transcriptional activation. Contrarily, less abundant, lower affinity binding sites could lead to only transient occupancy by the TF, not allowing sufficient time for activation, and instead marking them for inactivation by recruitment of histone deacetylases. In this manuscript we decided to focus on GATA6-NANOG-SOX2 binding dynamics. In a follow-up study, we are planning to use RIME and/or BioID to identify mechanisms and coregulators that may provide clues about how GATA6 functions as an activator/repressor. We tried to specifically address the reviewer’s question regarding an overlap between GATA6 and NANOG motifs within Epi CREs that could point to a more direct repression mechanism. Since the NANOG motif is not well defined, we

used the SOX2 recognition motif, and found that the two motifs do not overlap frequently (~2%). We have now added to this section that although Epi CREs contain motifs for both GATA6 and SOX2, they do not overlap often.

3. In order to provide more direct evidences supporting that GATA6 directly represses Epi-specific TFs by binding to and silencing their associated CREs, the authors could use CRISPRs to delete/mutate GATA6 binding motifs within some of those CREs. If GATA6 binding to those CREs is important for their silencing, then the deletion of GATA6 sites should reduce silencing.

4. Another interesting observation is that pluripotency TFs transiently bind to PrE CREs bound by GATA6. However, it is rather unclear whether the binding of the pluripotency TFs to those CREs is of any functional relevance. Are these TFs repressing, activating or simply binding to those CREs?. As suggested in the previous point, the authors could delete/mutate bindings sites for some of these pluripotency TFs in order to address their functional relevance.

Response to points 3 and 4.

The reviewer suggests an interesting experiment, which we had also considered when we made our initial observation of GATA6 and NANOG/SOX2 co-binding. To identify potential CREs where we could perform CRISPR deletion/mutations, we first looked at the distribution of GATA6 and SOX2 binding sites. While the binding sites of these two TFs do not overlap, as stated in point 2, there are multiple binding sites for these TFs within each CRE. We reasoned that removal of one motif could result in compensation by neighbouring motifs which would make interpretation of the experiment difficult and unclear and would thus be of little significance. To circumvent this issue we would need to substitute an entire CRE (400-1000bp) by an altered version containing scrambled versions of the TF motif in question. Furthermore, as most genes are controlled by more than one CRE this would imply several substitutions in one cell line to assess changes in expression of a single gene. Therefore, although we agree that this would be the best experiment to test our hypothesis, we deemed it too impractical. Therefore, while we develop alternative strategies to test this in future studies (namely rapid degron systems to allow controllable rapid protein depletion) we thought that testing if our co-binding observation also occurs *in vivo*, in blastocysts, would hold more biological significance. In our initial submission, we only had GATA6 CUT&RUN performed in early blastocysts. We have now performed NANOG CUT&RUN in early blastocysts, as well as GATA6 CUT&RUN in late blastocysts. This data is included in a new figure (Fig. 5). Altogether our data suggests that even in blastocysts, GATA6 and NANOG co-bind at Epi and PrE CREs. It will be exciting to determine the physiological relevance of this unanticipated observations in a follow-up study.

5. The authors show binding of GATA6 and Epi-specific TFs to a large set of shared CREs. Do the authors think that all these TFs simultaneously bind to the same CREs within the same cells and alleles? It is possible that, upon GATA6 overexpression, there is increased cellular heterogeneity, with cells expressing variable levels of GATA6 and other TFs. Therefore, the authors could perform sequential ChIPs (reChIP) to more conclusively show whether GATA6 and pluripotency TFs simultaneously bind to PrE and Epi-specific CREs during early induction time points.

We agree with the reviewer that cellular heterogeneity could contribute to the observed GATA6-NANOG co-binding at Epi and PrE sites. We were also concerned about heterogeneity of our *in vitro* differentiation system, and to assay for how much variation we see at each time-point during differentiation, we performed immunofluorescence (as mentioned below) which we have now added to the manuscript (Fig. S1). We also added to the study a different approach, where we performed RNAseq and ATAC-seq in unsorted populations and compared the transcriptional and chromatin accessibility changes with that of cells sorted for dsRed2 expression (expressed from the Dox-inducible dual promoter controlling both GATA6 as dsRed2 expression). We saw very little difference in RNAseq and ATACseq profiles when comparing bulk and sorted populations, which supports the homogeneity of the *in vitro* differentiation (Fig S1). The concern about using unsorted cells was specifically raised by reviewer 3, and a more detailed description of our finding is described in the response to reviewer 3 (point #1).

Even then, as the reviewer rightly points out, our CUT&RUN data would not be able to distinguish if GATA6 and NANOG bind the same allele or different alleles. As the reviewer suggested, we have performed reChIP for GATA6 (FLAG) and NANOG to determine if the two TFs indeed co-bind. We first ChIPed chromatin from cells at the 0h (to serve as control) and 2h time-points with FLAG, and reChIP'ed the eluted FLAG-bound chromatin with NANOG antibody. We have represented this data in Figure 5. We observed NANOG reChIP signal enriched at FLAG-bound regions supporting that GATA6 and NANOG indeed co-bind Epi as well as PrE-CREs on the same allele.

Minor points:

1. The authors should investigate GATA6 protein levels by western blot and/or immunofluorescence at various time points after induction.

We have profiled GATA6 levels early during in vitro differentiation by immunofluorescence and add to Fig S1.

2. The authors claim that at many PrE-specific regulatory elements, increased accessibility precedes transcriptional changes of the target genes. However, gene expression was investigated by polyA RNA-seq. The authors should investigate for at least some genes nascent transcripts rather than mRNA in order to better support their claims.

We agree that although our claim is plausible, polyA-RNAseq maybe be insufficient to substantiate it. As this conclusion is not essential for the main message of our study we have rewritten our interpretation as 'remodeling of chromatin landscape begins rapidly following GATA6 induction' in the main text.

3. The lack of H3K27me3 and H3K9me3 at PrE-specific CREs activated by GATA6 could be perhaps explained by the overall low levels of these histone marks in ESC cultured under 2i conditions. Moreover, these histone marks are not prevalent at distal CREs. The authors could use ChIP-seq (or similar) data generated in ICM for these histone marks to better support their findings.

We took the reviewers suggestion into account and obtained publicly available data generated from ICM cells using ChIP-seq (H3K27me3: SRX1617444, SRX1617445; H3K9me3: SRX2763443, SRX2763444). We plotted this data at the clusters of Fig. 2B. As the heatmap alongside shows, like our CUT&RUN data from uninduced ES cells, ICM ChIP-seq data shows similar lack of enrichment at PrE CREs. As the reviewer rightly points out, distal CREs can be devoid of repressive histone marks. We also plotted the ICM chip-seq data at coordinates of PrE genes used in Fig S2C (heatmap below). This also showed little enrichment at PrE genes in ICM cells. Since data from ICM cells did not show anything different from our CUT&RUN data for H3K27me3 and H3K9me3, we have not included these heatmaps in our revised manuscript. We would like to point out that, even though H3K27me3 appears to be relatively depleted at PrE CREs, a few loci did in fact have some enrichment for this histone mark in uninduced es cells and was completely depleted upon differentiation into PrE. An example of this locus is Sox17, which is depicted in FigS2B.

Reviewer #2

In this manuscript, Thompson et al. use GATA6-driven differentiation of mES cells to examine plasticity in cells from the inner cell mass. Specifically, the authors are interested in PrE vs Epi cell fates and how cis regulatory element regulation is altered through chromatin accessibility and factor localization to drive these cell fates. The authors perform multiple genomic assays (RNAseq, ATAC-seq, CUT&RUN, CUT&Tag, and Hi-C) to assess the environment at these cis regulatory elements and ultimately put forth a model on how NANOG and GATA6 act in this cell fate decision. Importantly, the authors also perform a few experiments in blastocysts to support their findings. Overall, the insights gained from the manuscript are exciting, and would be of interest to the field, however I have the following concerns:

1) In some of the datasets, it is unclear what the authors are presenting: for example, in Figure 1D, are the accessible locations called from the 48 hour timepoint, the 0hr timepoint, or all the accessible regions in any time point? Pending how these were called, for example, if they were called based on 0hr vs 48 hr, then are there locations that are up/down in the other time points that are not represented?

a. Related, the authors show in S1 non Epi or Pre-specific locations with changes in accessibility. What are these locations and why would they be changing upon GATA6 transgene expression?

We have now expanded our methods section to better describe how the analysis for Figure 1D was performed. We first combined the ATAC peaks identified at 0h and 48h. This union of 0h and 48h ATAC peaks was then associated with unique genes using GREAT (great.stanford.edu/public/html/; parameters: single gene, within 50kb). The gene-associated peaks were then filtered to only contain PrE- and Epi-specific genes, which we identified using scRNAseq data from E3.5 and E4.5 blastocysts. These ATAC-peaks were then classified as stable, increasing or decreasing using DiffBind by comparing 0h and 48h. We decided to use a comparison between 0h and 48h because

this yielded the most differential ATAC-peaks. Moreover, there were very few transient ATAC peaks that changed accessibility only during short periods of the differentiation process and not between 0 and 48 hours. Therefore, we believe that the heatmap we show in figure 1D is the most efficient representation of how chromatin accessibility changes during differentiation. Adding all intervening transient changes would be counterproductive as it would lead to many clusters of very few regions and dilute the strength of our observations.

a. We believe GATA6 is indeed responsible for initiation of most of the changes in accessibility shown in Fig S1. The heatmap shown here shows GATA6 cut&run signal at most sites undergoing accessibility changes, both at sites that increase and that decrease accessibility. It is true that not all these sites are associated with genes differentially expressed in PrE and Epi lineages in vivo. Since GATA6 is a transcription factor involved in gene regulation of multiple lineages other than PrE (e.g. heart, gut), Dox-induced GATA6 expression could result in GATA6 binding at many locations other than Epi and PrE. Likewise, NANOG binds over ten thousand genomic locations, of which only a subset are associated with the 200 genes that were defined as epiblast-specific according to blastocyst scRNA-seq datasets. It is precisely this difference between these two phenomena, 1) binding without affecting gene expression and 2) binding to control the expression of cell-type specific genes, that motivated us to perform a separate analysis on the genome and on Epi and PrE-specific genes. As our current analysis, showing ATAC signal at GATA6 binding sites also shows a very similar message, that GATA6 induces loss and gain of accessibility, we decided to not include one more plot as it could distract from the main observations.

2) The authors often overstate their data, drawing conclusions that are models or hypotheses based on the data. Below, I am listing a few examples, but there are more throughout the manuscript.

a. For Figure 1, it is surprising the authors do not just show the accessibility over time of the genes that are up and downregulated and then correlate these findings with the expression data. This is especially important as the authors make a claim on page 6 line 99 “changes in chromatin accessibility were more striking than those observed at the transcriptome level”. This is an extremely subjective comment that is not grounded in presenting quantitative results. Does striking mean number of peaks? If so, this should be explicit and also demonstrated in a quantitative manner.

b. on page 6 line 111 “...possibly due to a change in binding of a Sox7 repressor. This is a good example of why not all PrE CREs gain accessibility as it may have been expected”. While the authors are pulling Sox7 as an interesting example, there is no mechanistic experiment demonstrating the repressor binding has changed, and therefore this is not an actual mechanistic example, but rather a qualitative finding that could match multiple mechanisms.

c. On Page 7 line 113-114 “These data demonstrate that GATA6-driven chromatin remodeling is initiated immediately following induction”. The authors have not shown that GATA6 drives chromatin remodeling, but rather transgene expression of GATA6 results in alter accessibility. Some of these changes may be indirect and a result of other changes occurring during differentiation.

a. Our primary reason for first defining transcriptional changes, and then looking at the chromatin landscape at these genes, was to focus on changes of genes that form the Epi and PrE network in blastocysts. Our main goal was to understand what chromatin changes occur at genes that gain or lose expression when the PrE fate is specified in vivo. Since we had access to publicly available scRNAseq generated from early (E3.5) and late (E4.5) blastocysts, we compared transcriptional changes during our in vitro differentiation assay to the scRNAseq data, and defined genes showing the same trend of transcriptional changes in both datasets. We believe that for the question we chose to address this approach makes more sense than to characterize all of the changes that happen in mES cells once GATA6 is induced. The reviewer has rightly pointed out a comment that we agree is subjective. We have rewritten this sentence to “changes in chromatin accessibility were detected, (Fig S1C, S1D, S1E, S1F), indicating that remodeling of chromatin landscape begins rapidly following GATA6 induction”.

b. We agree that our statement regarding a repressor binding at Sox7 is mechanism which we did not test experimentally. We have therefore rephrased this sentence to not mention repressor.

c. While we think the reviewer’s comments about chromatin changes being indirect are valid, we would like to provide an explanation for why we think that is unlikely to be the case. Within 2hrs post Dox-induction, we detected very few differentially expressed genes. In particular, GATA6 downstream transcription factors (GATA4, SOX17, or any other PrE-specific factor, or known chromatin remodeler only become upregulated by 4-8 hours. Since

chromatin accessibility changes occurring within 2hrs are centered on GATA6 motifs at GATA6 peaks, we believe this strongly supports that chromatin remodeling is mediated by GATA6 binding, and not indirectly by another factor induced in response to the transgene expression. We have now included this explanation in our revised manuscript.

3) Throughout the paper, the authors show a lot of their data qualitatively. It is never clear how many replicates were performed, the overlap in those replicates (in terms of correlation), correlation in the timepoints over GATA6 induction, and quantitative peak calling/intersecting peak information (which could only be performed with the appropriate negative controls). Some of this information is included in the S1 supplement, but overlapping peaks from different datasets and making it more clear within the manuscript would benefit readers.

We show correlation between replicates from all CUT&RUN and CUT&TAG data used in our analysis in Fig. S8. PCA plots from RNAseq (S1D) and ATACseq (S1J) exhibit the variation within replicates used for these assays. We have now written the number of replicates used in each assay in the methods section. We apologize that we didn't properly describe the control used in our CUT&RUN assays. We did perform IgG CUT&RUN (rabbit anti-rabbit) alongside at each time-point, and pooled sequencing results from all IgG CUT&RUNs to generate a control that was used in peak calling. We have also included a peak-overlap between replicates for CUT&RUN performed for GATA6, NANOG, SOX2, GATA4, AND SOX17 in Table S1. Since we did not use peaks for histone marks anywhere in our analyses, we have not shown overlap between peaks from CUT&TAG assays. However, correlation between replicates from CUT&TAG assays are depicted in Fig S8. Additionally, we have now included the peaks used in all our heatmaps in table S1.

4) Experimentally, the authors perform no negative controls for their extensive CUT&RUN/CUT&Tag experiments (either IgG or no primary antibody). This is concerning, as we do not know the background cutting in their experiments.

Unfortunately, we failed to clearly describe the negative controls we used, in our first submission. We had performed IgG CUT&RUN in parallel with our CUT&RUN experiments, and this was used to call peaks and in downstream analysis. However, we overlooked mentioning that we performed IgG CUT&RUN in our methods. We have now revised our heatmaps to show background signal arising from IgG CUT&RUN. We also performed IgG-CUT&TAG multiple times but always failed to generate enough library for sequencing, indicating that there was very likely minimal background transposition.

5) Relatedly, the profiles presented for some of the datasets are concerning. For example, in Figure 2B, the background cutting/transposition events in the same profiling groups, at different timepoints are very similar. While there may be some similarities in the background, these are alarmingly similar. Without a negative control, this is especially concerning. Further, without access to the data to analyze, reviewers cannot confirm these findings.

We have now added background cutting emerging from IgG CUT&RUN to our heatmaps for direct comparison. Our data, including IgG controls, was uploaded to GEO (GSE181104) and has been publicly available since July 30, 2021. We provided this information to the editor but unfortunately, it was not relayed to the reviewers.

Minor comments:

1) In the presentation of Figure 1, the authors demonstrate genes that are up and downregulated over their timecourse of GATA6 induction. Then the authors examine CREs for PrE and Epi. As CREs typically mean promoters and other more distal regulatory elements, such as enhancers or silencer, how were these distal regulatory elements identified?

We used ATAC-peaks identified at 0h and 48h time-points for the analysis shown in Fig 1D and then GREAT to associate ATAC-seq peaks to genes as described in the methods section. We first combined the 0h and 48h ATAC-peaks to generate a master list. Each region in the list was then associated with unique genes using GREAT analysis (<http://great.stanford.edu/public/html/>; parameters: single gene, within 50kb). The gene-associated peaks were then filtered to only contain PrE- and Epi-specific genes which we identified using scRNAseq data acquired from E3.5 and E4.5 blastocysts. Following this, we used ATAC-peaks identified as differential when comparing 0h and 48h, using DiffBind, to finally get a list of ATAC-peaks in each category depicted in Fig 1D. This information is now more explicit in the methods and results section.

2) The K4me3 data presented in Figure 2B is very peculiar. It is perhaps due to these mostly being gene distal CREs, but if that is the case the authors should clarify. K4me3 over promoters should look much more consistent

with prior findings of this pattern, and the authors might consider showing those data to demonstrate the experiment worked.

We agree with the reviewer that the K4me3 pattern in Fig2B is not what is typically seen at active gene promoters. Regions in 2B are centered on GATA6 binding sites which we further classified as proximal and distal CREs, which typically do not coincide exactly at promoters. However, we have plotted K4me3 centered at gene promoters (Fig 3E, and S2C), which exhibit the expected K4me3 distribution pattern. Moreover, the browser shots (Fig 2C and S2B) also show deposition of K4me3 at PrE gene promoters that become active during in vitro PrE differentiation.

3) In Figure 2D it is surprising that there is no enrichment at all in the 0hr time point for GATA6 or K27ac, as free pA/G-MNase or Tn5 would cut/transpose to permit fragment release. This could be clarified by having the appropriate negative controls performed in parallel to the experiments.

We agree, and we had the same concern that we see very little background cutting/ transposition. Since we had performed IgG CUT&RUN in parallel with our CUT&RUN experiments, we concluded that GATA6 cut&run at 0h resembles IgG cut&run and shows very little background cutting. To confirm that NANOG-SOX2-GATA6 co-binding at PrE genes was not an artifact of CUT&RUN, we performed NANOG ChIP-seq as an orthogonal approach which confirmed our observation.

4) It is lost on me how in Figure 2E, left panel, these are accessible peaks, when the 0hr time point has occupancy of nucleosomes, similar to the 0hr time point on the right (non-accessible) panel.

We understand that this discrepancy can be confusing and was also pointed out by reviewer#3. GATA6 peaks are on an average 400bp in length and enrichment of ATAC signal does not mean that there are no nucleosomes across those 400bp. It just means that they are spread out and DNA is more easily accessible than in other genomic regions; fig 2E shows the nucleosomal signal of ATAC centered on gata6-motifs within that accessible region. Thus, it is possible that although the chromatin region is accessible, the GATA6 motifs themselves which are ~6bp long, tend to be protected/occluded by nucleosomes. As we describe, GATA6 motifs contained within 0h accessible sites, as well as sites opened only upon GATA6 induction, are occupied by nucleosomes in ES cells. The nucleosomes are remodeled only upon GATA6 expression, supporting that it functions as a pioneer TF. We have substantially rewritten this explanation to make it clearer to readers.

5) For Figure S2, the authors perform motif finding for peaks underlying the GATA6 locations. It is surprising that a) the authors show only one enriched motif and b) GATA6 motif was not identified. Were other significant motifs found? Further, later from the findings, we might have expected to see Oct-Sox-Nanog motifs; were they present?

The reviewer raises an interesting point. We provide here, an explanation as to why the GATA6 motif (and OCT4-SOX2-NANOG) is not identified as enriched. To understand why some GATA6 target peaks are occupied only later during differentiation, and not immediately following GATA6 induction, we compared 'late' GATA6 target sites, to 'early' sites and looked for motifs enriched specifically in the late category. Since the GATA6 motif is present in both early and late target sites, it would not be identified as 'enriched' in one category over the other. We have added this explanation in our main text. Since SOX2-NANOG bind PrE-sites only early in differentiation, these are also not expected to be found enriched specifically in late GATA6 targets.

6) On page 12 line 201-202 the authors state "Since Nanog levels decreased faster than Sox2 during PrE differentiation...". While this may be true, the slope for Nanog does look steeper than Sox2, this is not quantified and should be if the authors are going to make this statement. Also, given Nanog is more highly expressed, it should be normalized for expression.

We understand the reviewer's concern about the comment regarding Nanog levels decreasing faster than Sox2. While the reviewer is correct in saying that at the transcript level the decay in Nanog and Sox2 levels appear comparable, at the protein level (Fig S3A) SOX2 clearly persists much longer than NANOG. We now point out to the western blot (Fig3B) to better support the statement.

7) When the authors profile NANOG and SOX2 occupancy over GATA6 induction, they show that SOX2 binding persists while NANOG occupancy is decreased/lost. Did the authors discuss why SOX2 might be persistent in binding?

We think that persistent SOX2 protein levels at 8hr contributes to its slower loss in binding (diminished binding is apparent in Fig3D) even though NANOG binding is lost. We included an explanation of why SOX2 binding is diminished only slightly.

8) For Figure 4A, the second and third rows have the same number of locations listed (9830). It is surprising to me that this is an exact match, and therefore I think this is a typo.

We thank the reviewer for pointing this out. It was indeed a typo which we have now corrected.

9) The authors performed blastocyst CUT&RUN and it is not clear if they used individual blastocysts or pooled blastocysts.

We have clarified in our methods section that blastocyst CUT&RUN was performed in pooled blastocysts.

Reviewer #3

The manuscript by Thompson and collaborators reports ES cell transdifferentiation upon ectopic induction of Gata6 expression. They focus on the dynamic changes of transcription factors bindings on potential control regions. Such kind of experiments has previously been already carried out by Wamaitha et al (2015), but the present analysis is more thorough and, importantly examines early time points. They found that chromatin changes occur as early as 2 hours after transgene induction. The binding to PrE CREs shows different dynamics (early, late) revealing different mechanisms. GATA6 also binds to Epi CREs, as previously shown (Wamaitha, 2015). Here, a systematic analysis of Epi specific transcription factors (TFs) binding such as NANOG, SOX2 was carried out in parallel. A high number of Epi CREs bound by Epi TFs are also bound by GATA6. This enables to show the dynamics of eviction of these factors from Epi CREs by GATA6. Interestingly they observe a transient de novo binding of Epi TFs on PrE CREs occupied by GATA6 most likely due to novel open chromatin sites. They observe a difference in density and affinity for binding sites for GATA6 or for SOX2/OCT4 between Epi and PrE CREs that could explain the differences in binding dynamics. Altogether this is a thorough CRE binding analysis during GATA6 conversion of Epi cells into PrE, prolonging the findings by Wamaitha and collaborators. There are several concerns about the manuscript:

1- there is no mention of cell heterogeneity, which is present in both embryo and cells. Several groups have shown that NANOG and GATA6 proteins have cell-to-cell heterogeneous expression in E3.25 embryos (Dietrich and Hiiragi, Development 2007; Bessonard et al, Development 2014; Saiz et al, Nat Com 2016), and this is obvious in fig S4C. NANOG heterogeneity in ES cells has been reported by many labs, even cultured in 2iL (Abranches et al, Development 2014, 10.1242/dev.108910). Cell heterogeneity was observed until at least 8h after induction using the same dox system (Wamaitha, 2015; Schroter, 2015). The bulk RNAseq comparison with scRNAseq from embryonic cells is informative but a single cell analysis would have provided much more information on the cell states analysed. While transcription factors binding in single cells is challenging, scATAC-seq is being used increasingly. Authors have rightly used cell sorting with PECAM1 and PDGFRa, however only at 48h after induction. Using the FACS at each time point would be unable to regroup and analyse cells with a similar identity, as was done for example by LoNigro et al, Stem Cell Rep 2017 (10.1016/j.stemcr.2016.12.010). Trying to eliminate cell heterogeneity in samples (or use sc technologies) is important. For example, the binding of Epi TF and GATA6 at the same CREs could be due to cell heterogeneity. While this will be difficult to solve, this limitation should be at least commented in the main text.

We would like to provide a detailed explanation of the reasoning behind our approach for characterizing early time-points in unsorted populations. At the onset of our in vitro study, we were also concerned about cellular heterogeneity, and to assay for how heterogeneous our in vitro differentiation was, we performed bulk-RNAseq and bulk-ATACseq in cells sorted for dsRed2 expression (expressed from the Dox-inducible dual promoter controlling both GATA6 as dsRed2 expression) and unsorted populations. These figures were removed from the initial submission based on input from our peers who found that the comparison of sorted and unsorted was distracting from the main message of the paper. We have now added these data (Fig S1). We observed minimal differences in expression changes and chromatin accessibility dynamics upon comparison of sorted and unsorted populations. This allowed us to confidently proceed with CUT&RUN and CUT&TAG assays in unsorted populations. Moreover, genes like *Pdgfra*, that could effectively sort out PrE-like cells, are transcriptionally abundant only starting at 4hr, making sorting possible only after 8 hours. Since our primary goal was to investigate transcription-factor dynamics at early time-points, we used bulk-population without sorting. We have added immunofluorescence for GATA6 at early time-points to show that our bulk-population is indeed homogenous in expression (Fig S1C). We agree that our data from embryos staged at E3.5 represent findings from heterogeneous cells. scATACseq is a very good

suggestion and is something we are working towards. However, we believe that it is beyond the scope of the current paper. It would also require a large financial investment and not necessarily alter the main conclusions of this study. As described under reviewer 1 point#5, we have now performed reChIP to particularly address if NANOG and GATA6 cobind at the same CREs. This data is included in the revised manuscript in the new figure 5 and supports our claim that NANOG-GATA6 indeed co-bind both Epi and PrE CREs on the same allele. However, we do acknowledge that no matter how homogenous our system may appear, there would still be some cellular variability. We have now mentioned this limitation in this subsection of our results.

2- ES cells are most similar to an E4.5 epiblast cells and ES cells do not give rise to PrE cells in vivo. So what is induced here is transdifferentiation. Due to different starting cell states and chromatin landscapes, the mechanism to produce PrE cells is certainly different than in the embryo. ES cells cannot be used to study early time points of PrE differentiation. The analysis that was carried out on embryos is thus important (although difficult to interpret due to cell heterogeneity). Nevertheless, it should be reminded in the main text that this artifactually induced transdifferentiation is certainly different than in vivo.

We agree that transdifferentiation is a better term for what we describe here. Due to the intrinsic heterogeneity of this cell fate decision, it is very challenging to profile minute temporal changes that accompany PrE or Epi lineage commitment. TF dynamics especially at the single-cell level, is hard to study in embryos, given the limitations of currently available techniques. For this reason, we characterized the in vitro time-points that more closely resemble the different blastocyst cell-populations and used these time-points to identify novel regulatory mechanisms. However, we do agree that ES cell to PrE differentiation is a transdifferentiation system, and we have now added this to our introduction and the results section where we describe our system. Importantly, we also expanded our experiments in vivo to include more timepoints for GATA6 binding, and we also profiled NANOG binding which validated our in vitro observations. These new observations made us more confident that our transdifferentiation experiment is a good starting point to study gene regulation by TFs in blastocysts.

3- It is interesting to see that Gata6 RNA levels reach their maximum already after 2 hours of induction. However, are these levels (transgene and endogenous) physiological?. What are these levels compared to in vivo.

The reviewer raises an interesting point. We did verify that endogenous and transgene Gata6 levels were similar (Fig.2A). However, it is indeed possible that transgene induction would raise both these types of Gata6 mRNAs beyond physiological levels. Unfortunately, It would be very hard to directly compare expression levels of genes in vitro to blastocysts. First, more than 95% of the cells become PrE in vitro, while only 1/3 of blastocyst cells are PrE. Thus, RNAseq or cDNA data quantified from whole blastocysts would not be comparable to an in vitro PrE system. scRNAseq from blastocysts could be an alternative, but again the difference in availability of material as compared to ES cell differentiation system could result in a much lower number of 'good quality' reads per gene making the comparison of exact levels with in vitro differentiation unfair. Furthermore, scRNAseq is not great at absolute quantifications. Comparison of our bulk-RNAseq with scRNAseq (Fig 1) shows that in vitro differentiation follows a similar trend to PrE differentiation in vivo, however the absolute values would not be directly comparable. As we wanted to model PrE differentiation as closely as possible, we resorted to a pulse of Doxycycline addition rather than continuous exogenous GATA6 expression which we thought would resemble what happens in embryos.

4- GATA4 and SOX17 relay GATA6 on PrE CREs. Is it the case as well on Epi CREs?

Upon in vitro GATA6 induction, we found that GATA4 and SOX17 expression increased between 4 – 8 hours, and thus rationalized that their binding could be profiled only at later time-points. At these time-points, GATA4 and SOX17 were found to bind GATA6 bound-PrE sites, but not at Epi sites (heatmap on the right). We decided to not focus on GATA4-SOX17 at Epi CREs, because we didn't detect any binding at these sites. We have included a line stating our rational in the revised manuscript and in addition, we could add the heatmap below to the draft if it would be valuable to readers. We believe that the different timings of GATA4/SOX17 expression and GATA6 binding at Epi CREs speaks against such inclusion.

5- Are there some of the CREs described here that were previously validated functionally (i.e., indeed control the target gene). This would enrich the data.

While the reviewer's comment is important, we think that there are very few CREs functionally validated in vitro or in vivo, to allow a fair comparison between validated CREs and the CREs we have bioinformatically identified in our study using ATAC-seq and TF-binding data. Moreover, CREs controlling genes specifically during PrE commitment are not extensively characterized. However, functionally validated CREs controlling Epiblast genes (e.g, *Sox2*, *Nanog*), which lose activity during PrE differentiation, were identified in our study. We however do agree that our study would benefit if the CREs we identify, were also identified by other studies. To address this, we have tried two approaches. First, we looked at the vista enhancer database, and determined which of the CREs identified in our study are also classified as VISTA enhancers. We found few overlapping CREs. This low extent of overlap can be explained by the fact that CREs regulating genes expressed in endodermal lineages (and therefore more likely to be similar to PrE enhancers) are underrepresented in the VISTA database as compared to CREs regulating genes expressed in ectodermal and mesodermal lineages. As a second approach, we determined if our CREs were classified as putative enhancers in the ENCODE CRE database and found a large overlap between both datasets. We have included information on which of our CREs intersect with ENCODE CREs in table S1.

6- Fig 2E left panel: why does it look like a nucleosome is present at 0h while it is supposed to be accessible chromatin?

This concern was also raised by reviewer #2 (minor point 4). GATA6 peaks are on an average 400bp in length and enrichment of ATAC signal does not mean that there are no nucleosomes across those 400bp. It just means that they are spread out and DNA is more easily accessible than in other genomic regions; fig 2E shows the nucleosomal signal of ATAC centered on *gata6*-motifs within that accessible region. Thus, it is possible that although the chromatin region is accessible, the GATA6 motifs themselves which are ~6bp long, tend to be protected/occluded by nucleosomes. As we describe, GATA6 motifs contained within 0h accessible sites, as well as sites opened only upon GATA6 induction, are occupied by nucleosomes in ES cells. The nucleosomes are remodeled only upon GATA6 expression, supporting that it functions as a pioneer TF. We have substantially rewritten this explanation to make it clearer to readers.

7- Lists of ranked genes/CREs should be provided for the different experiments (Figures 1C/S1D; 2B; 3B; 3E; 4A, 4F, 5B bottom part). As well, I guess that all data will be deposited to a publicly available repository, as nothing is mentioned in the manuscript.

We thank the reviewer for pointing this out. We added a list of ranked CREs used in the different heatmaps to the supplementary table, with tabs labelled as the figure. All data were deposited onto GEO (GSE181104) and made public on July 30, 2021. Unfortunately, it appears that this information was relayed to reviewers.

REVIEWER COMMENTS

Reviewer #1 (Remarks to the Author):

The authors have successfully addressed most of my previous concerns. However, I am not fully convinced by their arguments regarding my major concerns #3&4:

3. In order to provide more direct evidences supporting that GATA6 directly represses Epi-specific TFs by binding to and silencing their associated CREs, the authors could use CRISPRs to delete/mutate GATA6 binding motifs within some of those CREs. If GATA6 binding to those CREs is important for their silencing, then the deletion of GATA6 sites should reduce silencing.

4. Another interesting observation is that pluripotency TFs transiently bind to PrE CREs bound by GATA6. However, it is rather unclear whether the binding of the pluripotency TFs to those CREs is of any functional relevance. Are these TFs repressing, activating or simply binding to those CREs?. As suggested in the previous point, the authors could delete/mutate bindings sites for some of these pluripotency TFs in order to address their functional relevance.

The authors could try to find Epi or PrE CREs harboring single TFBS for Gata6 or pluripotency TFs (e.g. Sox2/Oct4), respectively. Among the hundreds of identified CREs it is likely that some of them might harbor single sites for the investigated TFs. Even if the target genes are regulated by multiple CREs, the authors could still investigate the effects that the TFBS deletions might have on individual CREs by evaluating H3K27ac or eRNA levels for example. Another possibility is to investigate the candidate CREs using reporter assays, which would enable the authors to more easily modify the investigated CREs and delete or mutate multiple TFBS. I think that the binding of GATA6 to Epi CREs and of pluripotency TFs to PrE CREs are among the most interesting observations reported in the manuscript, but without any additional experiments it is currently unclear whether these binding events have any functional significance, even if they also occur in vivo.

Reviewer #2 (Remarks to the Author):

The authors have done an excellent job addressing all the concerns.

Reviewer #3 (Remarks to the Author):

The authors have addressed most of the points I raised, however a major concern remains about cell heterogeneity.

In vitro, re-ChIP experiments validate co-binding of NANOG and GATA6 on the same CREs.

However, this is more complicated in vivo. As I was indicating in my review, there is cell to cell heterogeneity already in E3.25 embryos (Dietrich and Hiiragi, Development 2007; Bessonard et al, Development 2014; Saiz et al, Nat Com 2016). Actually changes still need to be made on page1 line22, as the factors mentioned there are all present but expressed heterogeneously, and not uniformly as written.

This has serious implications as re-ChIP or sc analysis have not (or cannot) been carried out.

On the new manuscript authors are showing a novel analysis on E4.5 embryos. This is very

interesting and speaking. Indeed at this stage cells are well differentiated into Epi with no GATA6 presence or PrE with no NANOG presence (Plusa, Development 2008). Thus NANOG and GATA6 binding on the same CREs cannot occur in the same cells at this stage. However this is very interesting as this means that GATA6 remains on Epi CREs in PrE cells at E4.5 (same for NANOG on PrE CREs in Epi cells). Mixing ES cells at 0h (equivalent to E4.5 Epi cells) and ES cells at 24h (equivalent to E4.5 PrE cells) of Gata6 induction and analysing them as bulk would probably be equivalent as taking whole embryos at E4.5. Co-binding at earlier stages (E3.5) cannot be determined here. At this stage there are cells already engaged toward Epi, others to PrE and others that are still undifferentiated (Saiz et al, Nat Com 2016). Thus binding to the same CREs could be due to cell mixing like at E4.5 or possibly to co-binding. This needs to be explained clearly. Co-binding cannot be determined with the current experiments.

As in vivo undifferentiated cells, that are giving rise to Epi or PrE cells, are not in an Epi state like in the in vitro situation, the mechanism of differentiation might be different. For example the amount of NANOG and GATA6 proteins could be lower compared to in vitro and thus the quantity and quality of the occupied CREs could be different. This is why I was enquiring about relative levels of Gata6 in my review. Thus for this early step of differentiation ES cell transdifferentiation cannot be a model for the in vivo situation.

Concerning Point 4, I think it is important to mention that Sox17 and GATA4 do not bind to Epi CREs. This is interesting as it indicates that these proteins are not involved directly in the maintenance of Epi repression within PrE cells.

Reviewer #1

The authors have successfully addressed most of my previous concerns. However, I am not fully convinced by their arguments regarding my major concerns #3&4:

We thank the reviewer for their suggestions and are happy we addressed most of their concerns.

Comments from round 1

3. In order to provide more direct evidences supporting that GATA6 directly represses Epi-specific TFs by binding to and silencing their associated CREs, the authors could use CRISPRs to delete/mutate GATA6 binding motifs within some of those CREs. If GATA6 binding to those CREs is important for their silencing, then the deletion of GATA6 sites should reduce silencing.

4. Another interesting observation is that pluripotency TFs transiently bind to PrE CREs bound by GATA6. However, it is rather unclear whether the binding of the pluripotency TFs to those CREs is of any functional relevance. Are these TFs repressing, activating or simply binding to those CREs?. As suggested in the previous point, the authors could delete/mutate bindings sites for some of these pluripotency TFs in order to address their functional relevance.

Comments from round 2

The authors could try to find Epi or PrE CREs harboring single TFBS for Gata6 or pluripotency TFs (e.g. Sox2/Oct4), respectively. Among the hundreds of identified CREs it is likely that some of them might harbor single sites for the investigated TFs. Even if the target genes are regulated by multiple CREs, the authors could still investigate the effects that the TFBS deletions might have on individual CREs by evaluating H3K27ac or eRNA levels for example.

Another possibility is to investigate the candidate CREs using reporter assays, which would enable the authors to more easily modify the investigated CREs and delete or mutate multiple TFBS.

I think that the binding of GATA6 to Epi CREs and of pluripotency TFs to PrE CREs are among the most interesting observations reported in the manuscript, but without any additional experiments it is currently unclear whether these binding events have any functional significance, even if they also occur in vivo.

We agree that the wide-spread binding of GATA6 and NANOG to the same CREs is among the most interesting and unanticipated observations in our study. We also understand the reviewer's criticism that we have not functionally dissected the importance of these observations. To address this we have carefully revised the manuscript to ensure that we do not claim functionality to these observations. We also included in the discussion that the experiment suggested by the reviewer would address this issue. As detailed below, we agree that the experiments proposed by the reviewer are a good start to assess the potential functional significance of GATA6-NANOG cobinding. However, regardless of the outcome of these experiments, a rigorous functional investigation would still require much deeper investigation better suited for a follow-up study fully dedicated to this question. We believe doing these experiments as part of the current manuscript will provide an incomplete assessment, delay publication, and result in potentially incorrect, incomplete, or misleading mechanistic models. Furthermore, we argue that as it stands, our work and observations represent significant conceptual advancement to the field that should not be dependent on specific results from additional experiments involving genome manipulation.

Deletion of TFBS as the reviewer proposes above may provide functional insight but would require a disproportioned investment of resources compared to the probability that such experiments would provide potential mechanistic insights with certainty. First, to bypass compensation by other TFBS in the same CRE the reviewer suggests targeting Epi or PrE CREs with a single GATA6 and SOX2 binding site. However, it is likely that CREs with low density of SOX2/GATA6 motifs may not be the preferential targets of these TFs or that at these target genes SOX2/GATA6 play a smaller role in gene regulation of target genes. This goes along with the notion that TF motif density, can impact the degree of functional effect of TF binding on the CRE (proposed in our manuscript, and also by Buecker, Cell stem cell, 2014). Second, as redundant CREs frequently control the same gene (Osterwalder Nature 2018), this prevents the use of gene transcript levels to assess the effect of loss of activity from one specific CRE controlling that gene. The reviewer suggests a very interesting idea of using H3K27ac or eRNA levels as alternative readouts. However, we observed that reshuffling of TF binding precedes H3K27ac changes by at least 8 hours. Therefore, addressing the reviewer's request would require additional standardizations and tests comparable in effort, difficulty, and volume to the data already included in our current manuscript. Assessing eRNA levels is also not trivial. Although eRNAs tend to correlate well with enhancer activity this is not always true (Mikhaylichenko Gen Res, 2018). Third, we would need to generate several mES

mutant cell lines as it is likely that the effect of TF cobinding is CRE- and gene-specific and focusing on few loci could lead to incorrect conclusions. We do not believe these arguments should prevent us from performing these experiments and in fact we have initiated some of them. We argue instead that they are not trivial and to perform them will require a large time and resource investment better suited to a follow up study fully dedicated to this goal.

As the reviewer mentions, a faster alternative could be to employ reporter assays to test the effects of cobinding. However, episomal based assays will not represent the nuances of the endogenous sites where cobinding occurs. For example, histone dynamics would not be properly modelled in a plasmid-based luciferase system. These tools have been used to dissect the modularity of enhancer elements (Thomas Mol Cell, 2021; Zhou G&D, 2014) and to understand how TF binding regulates enhancer activity (Singh Gen Res 2021; Grossman PNAS 2016). However, these studies entirely focused either on dissecting the regulation of a single gene by an enhancer or groups of enhancers, or dissecting TF regulatory logic in general. In addition, they all complemented these assays with genomic deletions. To achieve the goal requested by the reviewer we must employ similar strategies that would require a large investment of time and resources and, like the studies cited above, are better suited as part of a stand-alone publication.

Our goal was to describe the remodeling of the genome when the PrE fate is determined. We present a highly resolved temporal characterization of TF binding, changes in histone marks, chromatin accessibility and contacts between enhancers and promoters. We agree that the most intriguing and unexpected observation is the widespread cobinding of TFs at Epi and PrE CREs. Following a suggestion from this reviewer we confirmed it with ReChIP-seq –a technique that has never been reported genome-wide for TFs– and tested if cobinding also occurs in vivo at different stages of blastocyst development. Importantly, our blastocyst TF binding data was generated using an improved protocol that substantially increases the robustness of the method compared to the only other dataset currently available (Hainer Cell 2019). Thus, we believe that in addition to our study, the datasets released upon publication of our work will themselves constitute a unique and valuable dataset to researchers in the field. Additionally, the observation that NANOG and GATA6 cobinding during Epi and PrE lineage commitment is not only novel but highly unexpected. We propose that it is linked to the phenomenon of shared regulatory networks driving multifurcation of cell lineages from common progenitors, which has been highlighted recently by several single-cell RNAseq studies (such as Mittnenzweig Cell, 2021).

In summary, while we agree that dissecting the functional significance of TF cobinding is important and modulating endogenous TF binding sites would be the best way to systematically test the function of cobinding, these set of experiments are an entirely new line of investigation. To report a short investigation into its function that would still fit within the limits of this paper would likely lead to wrong conclusions and models. Examples abound of important studies that describe new observations that open novel avenues for mechanistic exploration. For example, description of bivalent chromatin domains in 2006 initiated a large number of studies that significantly increased our understanding of the impact of epigenetic regulation of gene expression. However, the precise function of these domains in cell fate decisions remains unclear even now. We believe our study falls under this category. We are happy that reviewers 2 and 3 agree with us and hope that reviewer 1 would allow us to take a deeper dive into investigating this fascinating observation with all the care and patience it deserves in a follow-up manuscript.

Reviewer #2

The authors have done an excellent job addressing all the concerns.

We are very pleased that the reviewer considered we did an excellent job addressing all their thorough review.

Reviewer #3

The authors have addressed most of the points I raised, however a major concern remains about cell heterogeneity. In vitro, re-ChIP experiments validate co-binding of NANOG and GATA6 on the same CREs. However, this is more complicated in vivo. As I was indicating in my review, there is cell to cell heterogeneity already in E3.25 embryos (Dietrich and Hiiragi, Development 2007; Bessonard et al, Development 2014; Saiz et al, Nat Com 2016). Actually changes still need to be made on page1 line22, as the factors mentioned there are all present but expressed heterogeneously, and not uniformly as written. This has serious implications as re-ChIP or sc analysis have not (or cannot) been carried out.

We are happy that the reviewer considers we addressed the points they raised. We have rephrased our introduction to highlight the heterogeneity of TF expression in early blastocysts and deleted “uniformly”.

On the new manuscript authors are showing a novel analysis on E4.5 embryos. This is very interesting and speaking. Indeed at this stage cells are well differentiated into Epi with no GATA6 presence or PrE with no NANOG presence (Plusa, Development 2008). **Thus NANOG and GATA6 binding on the same CREs cannot occur in the same cells at this stage.** However this is very interesting as this means that GATA6 remains on Epi CREs in PrE cells at E4.5 (same for NANOG on PrE CREs in Epi cells). Mixing ES cells at 0h (equivalent to E4.5 Epi cells) and ES cells at 24h (equivalent to E4.5 PrE cells) of Gata6 induction and analysing them as bulk would probably be equivalent as taking whole embryos at E4.5. Co-binding at earlier stages (E3.5) cannot be determined here. At this stage there are cells already engaged toward Epi, others to PrE and others that are still undifferentiated (Saiz et al, Nat Com 2016). Thus binding to the same CREs could be due to cell mixing like at E4.5 or possibly to co-binding. **This needs to be explained clearly. Co-binding cannot be determined with the current experiments.** As in vivo undifferentiated cells, that are giving rise to Epi or PrE cells, are not in an Epi state like in the in vitro situation, the mechanism of differentiation might be different. For example the amount of NANOG and GATA6 proteins could be lower compared to in vitro and thus the quantity and quality of the occupied CREs could be different. This is why I was enquiring about relative levels of Gata6 in my review. Thus for this early step of differentiation ES cell transdifferentiation cannot be a model for the in vivo situation.

The reviewer raises important points that we apologize for not addressing properly in our revision. We added a paragraph to the discussion to correct this. We agree that at E3.5 we cannot distinguish between co-binding and mixing of Epi and PrE-specified cells and now explain this in the text. We also ensured that whenever referring to the in vivo situation we have written that NANOG and GATA6 bind at shared regulatory elements and not that they co-bind together to the same elements. We also used this new paragraph to re-iterate that although useful, this transdifferentiation model, is not a perfect recapitulation of in vivo differentiation. As mentioned by the reviewer we also added to our results section that our data suggests that at E4.5 GATA6 is still bound at Epi genes rather than cobound as the two TFs are not co-expressed in the same cells at this stage.

Concerning Point 4, I think it is important to mention that Sox17 and GATA4 do not bind to Epi CREs. This is interesting as it indicates that these proteins are not involved directly in the maintenance of Epi repression within PrE cells.

We have now added a heatmap to Fig S3 and described the results in the main text.

REVIEWERS' COMMENTS

Reviewer #3 (Remarks to the Author):

The authors have addressed my concerns appropriately.

REVIEWERS' COMMENTS

Reviewer #3 (Remarks to the Author):

The authors have addressed my concerns appropriately.

Thank you